# Unfolded protein response transducer IRE1-mediated signaling independent of XBP1 mRNA splicing is not required for growth and development of medaka fish

Tokiro Ishikawa[1], Makoto Kashima[2], Atsushi J Nagano[3], Tomoko Ishikawa-Fujiwara[4], Yasuhiro Kamei[5], Takeshi Todo[4], Kazutoshi Mori[1]*

[1]Department of Biophysics, Graduate School of Science, Kyoto University, Kyoto, Japan; [2]Research Institute for Food and Agriculture, Ryukoku University, Otsu, Japan; [3]Faculty of Agriculture, Ryukoku University, Otsu, Japan; [4]Graduate School of Medicine, Osaka University, Suita, Japan; [5]Spectrography and Bioimaging Facility, National Institute for Basic Biology, Okazaki, Japan

**Abstract** When activated by the accumulation of unfolded proteins in the endoplasmic reticulum, metazoan IRE1, the most evolutionarily conserved unfolded protein response (UPR) transducer, initiates unconventional splicing of XBP1 mRNA. Unspliced and spliced mRNA are translated to produce pXBP1(U) and pXBP1(S), respectively. pXBP1(S) functions as a potent transcription factor, whereas pXBP1(U) targets pXBP1(S) to degradation. In addition, activated IRE1 transmits two signaling outputs independent of XBP1, namely activation of the JNK pathway, which is initiated by binding of the adaptor TRAF2 to phosphorylated IRE1, and regulated IRE1-dependent decay (RIDD) of various mRNAs in a relatively nonspecific manner. Here, we conducted comprehensive and systematic genetic analyses of the IRE1-XBP1 branch of the UPR using medaka fish and found that the defects observed in XBP1-knockout or IRE1-knockout medaka were fully rescued by constitutive expression of pXBP1(S). Thus, the JNK and RIDD pathways are not required for the normal growth and development of medaka. The unfolded protein response sensor/transducer IRE1-mediated splicing of XBP1 mRNA encoding its active downstream transcription factor to maintain the homeostasis of the endoplasmic reticulum is sufficient for growth and development of medaka fish.
DOI: https://doi.org/10.7554/eLife.26845.001

*For correspondence: mori@upr.biophys.kyoto-u.ac.jp

Competing interests: The authors declare that no competing interests exist.

## Introduction

Essentially, all eukaryotic cells cope with endoplasmic reticulum (ER) stress, the accumulation of unfolded/misfolded proteins in the ER, by activating the unfolded protein response (UPR). This activation leads to maintenance of the homeostasis of the ER, where newly synthesized secretory and transmembrane proteins are folded and assembled (*Kaufman, 1999*; *Mori, 2000*). Among the three signaling pathways operating in mammalian cells (IRE1, PERK and ATF6), the most evolutionarily conserved UPR transducer is IRE1, a type I transmembrane-type protein kinase and endoribonuclease (*Ron and Walter, 2007*). Yeast and metazoan IRE1 activated by ER stress-induced oligomerization and *trans*-autophosphorylation initiates a splicing reaction and thereby specifically removes introns of 252 and 26 nucleotides from HAC1 and XBP1 precursor mRNA, respectively, which encode their respective downstream transcription factors (*Mori, 2009*). Because the HAC1 intron has the ability to block translation, only spliced HAC1 mRNA is translated in yeast cells (*Chapman and Walter, 1997*; *Kawahara et al., 1997*). In contrast, because XBP1 intron is too short

to block translation, both unspliced and spliced XBP1 mRNA are translated to produce the unspliced form of XBP1, designated pXBP1(U), and the spliced form of XBP1, designated pXBP1(S), in meta-zoan cells (*Calfon et al., 2002*; *Yoshida et al., 2001*).

pXBP1(U) and pXBP1(S) share the N-terminal region containing the basic leucine zipper (bZIP) domain but differ in their C-terminal regions (see Figure 6C). Since the C-terminal region of pXBP1 (S) contains a transcriptional activation domain (AD), pXBP1(S) functions as a potent transcription factor (*Yoshida et al., 2001*). In contrast, as pXBP1(U) binds to pXBP1(S) and targets pXBP1(S) to proteasomal degradation via a domain designated degron here when the cell recovers from ER stress, pXBP1(U) functions as a negative regulator of pXBP1(S) (*Yoshida et al., 2006*). It should be noted that XBP1 precursor mRNA is expressed at a low level under normal conditions but is induced during ER stress with a similar time course to ER chaperone mRNAs due to the presence of the same *cis*-acting ER stress response element in its promoter region (*Yoshida et al., 2000*). Thus, a higher level of pXBP1(U) is produced from transcriptionally induced XBP1 mRNA when IRE1 is inactivated due to recovery from ER stress. It was also reported recently that pXBP1(U) suppresses autophagy by promoting the degradation of FoxO1 (*Zhao et al., 2013*) (see *Figure 1A*).

Interestingly, IRE1 functions independently of XBP1 in two ways (see *Figure 1A*). First, the adaptor molecule TRAF2 binds to activated and phosphorylated IRE1 and sequentially activates ASK1 (MEKK1), JNKK and JNK, leading to phosphorylation and activation of c-Jun (JNK pathway) (*Urano et al., 2000*). Second, activated IRE1 cleaves various mRNAs relatively non-specifically, a process called regulated IRE1-dependent decay of mRNAs (RIDD pathway) (*Hollien and Weissman, 2006*). Although both are pro-apoptotic (*Han et al., 2009*; *Urano et al., 2000*), the physiological significance of these two pathways remains unclear.

Here, we adopted a genetic approach toward a comprehensive understanding of IRE1 function using medaka fish (*Oryzias latipes*), in which removal of the 26-nucleotide intron, which is exactly the same size as that in mammals, from precursor XBP1 mRNA is observed in response to ER stress (*Ishikawa et al., 2011*). The only difference is in the expression pattern of IRE1: although IRE1α is ubiquitously expressed and IRE1β is expressed only in the gut in mice (*Bertolotti et al., 2001*), both IRE1α and IRE1β are expressed ubiquitously in medaka fish (*Ishikawa et al., 2011*).

## Results

### IRE1α/β-double knockout (DKO) is detrimental to growth and development of medaka

We employed the targeting-induced local lesions in genomes (TILLING) method to identify IRE1α- and IRE1β-knockout (KO) medaka, as described in Materials and methods. The N-terminally truncated fragment of IRE1α (C156X) or IRE1β (Y164X) produced from the mutated allele must have lost its functionality as its transmembrane, protein kinase and ribonuclease domains are excluded (*Figure 1B and C*). We confirmed the presence of the expected mutation in IRE1α mRNA or IRE1β mRNA expressed in respective KO embryo at 5 days post-fertilization (dpf) by direct sequencing of RT-PCR products (*Figure 1D*). Both IRE1α- and IRE1β-single KO medaka were born and developed normally, and tunicamycin-induced splicing of XBP1 mRNA was observed normally in their embryos at 4 dpf; tunicamycin elicits ER stress by blocking protein N-glycosylation (*Figure 1E*). We therefore crossed the IRE1α-heterozygote with the IRE1β-heterozygote to obtain IRE1α/β-DKO medaka and found that tunicamycin-induced splicing of XBP1 mRNA was completely lost in the embryos of IRE1α/β-DKO medaka (*Figure 1E*). These results indicate that IRE1α and IRE1β are redundant in medaka, unlike in mice. It should be noted that all IRE1α/β-DKO medaka die within 2 weeks after fertilization regardless of their ability or inability to hatch (data not shown here, see Figure 10A; medaka usually hatch at 7 dpf and become fertile 2 months after hatching). Thus, complete absence of IRE1 functions is detrimental to the growth and development of medaka.

### XBP1-KO medaka exhibit a more severe phenotype than IRE1α/β-DKO medaka in the hatching gland

We employed the transcription activator-like effector nuclease (TALEN) method to generate XBP1-KO medaka, as described in Materials and methods. The N-terminally truncated fragment of XBP1 (Δ8) produced from the mutated allele must have lost its functionality as its bZIP domain is excluded

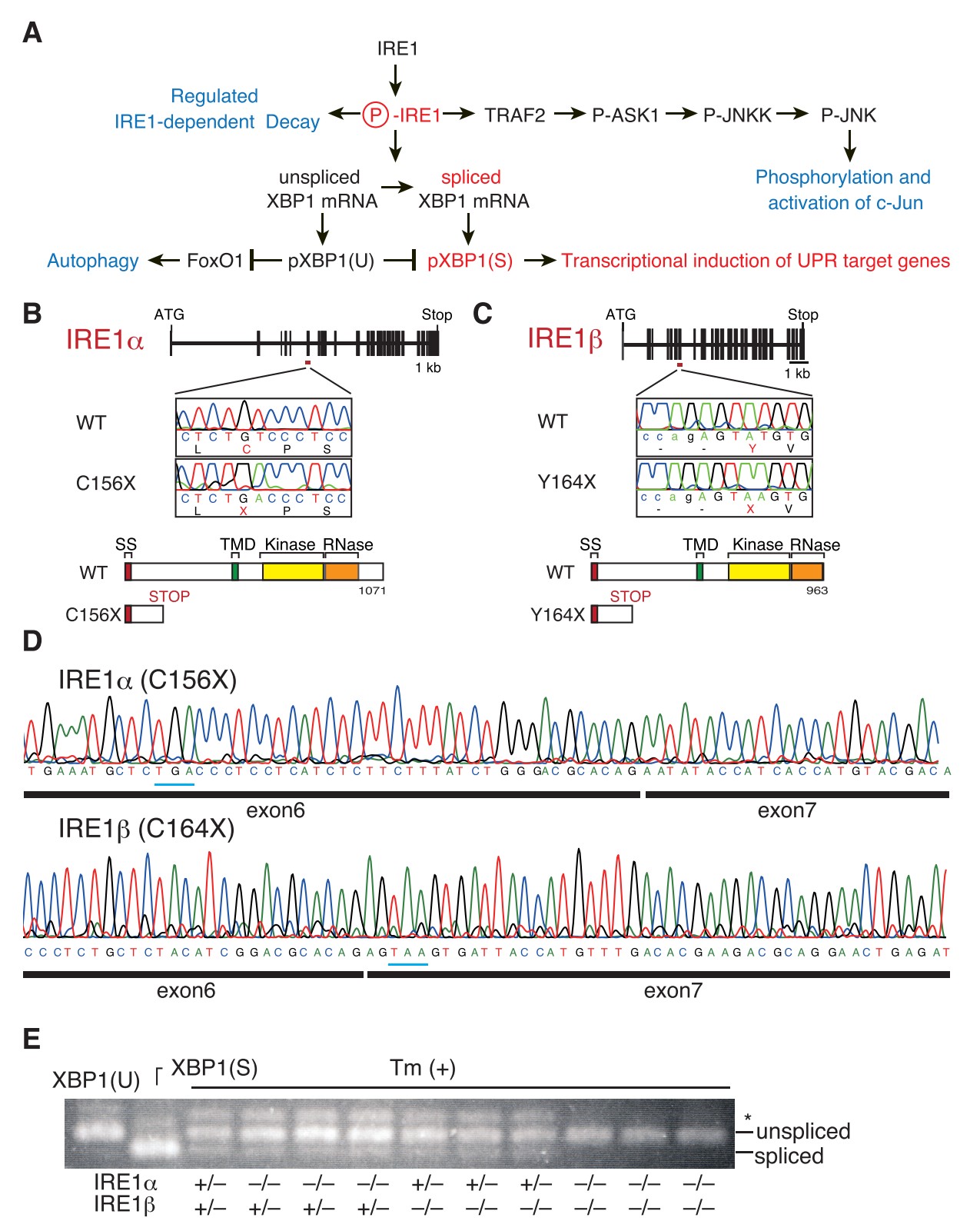

**Figure 1.** Construction and initial characterization of IRE1α-KO, IRE1β-KO and IRE1α/β-DKO medaka. (**A**) Schematic representation of the IRE1-mediated signaling pathways. (**B**) Schematic representation of the WT and C156X-mutant IRE1α genes and proteins. SS, TMD, Kinase and RNase denote the signal sequence, transmembrane domain, protein kinase domain and ribonuclease domain, respectively, which also apply to (**C**). (**C**) Schematic representation of the WT and Y164X-mutant IRE1β genes and proteins. (**D**) RT-PCR products corresponding to a part of IRE1α mRNA

*Figure 1 continued on next page*

*Figure 1 continued*

expressed in embryo at 5 dpf of IRE1α-KO medaka (top) and a part of IRE1β mRNA expressed in embryo at 5 dpf of IRE1β-KO medaka (bottom) were sequenced. Expected stop codons are underlined with blue. Note that they do not have an intron. (E) Total RNA prepared from embryos at 4 dpf of the indicated genotypes which had been treated (+) with 4 µg/ml tunicamycin (Tm) for 4 hr was subjected to RT-PCR. XBP1(U) and XBP1(S) indicate RT-PCR product corresponding to unspliced and spliced XBP1 mRNA, respectively. The asterisk denotes a heteroduplex of unspliced and spliced XBP1 mRNA.

DOI: https://doi.org/10.7554/eLife.26845.002

(*Figure 2A*). We confirmed the presence of the expected deletion in XBP1 mRNA expressed in KO embryo at 5 dpf by direct sequencing of RT-PCR products (*Figure 2B*).

Close examination of embryos expressing EGFP under the control of the ER chaperone BiP promoter (P$_{BiP}$-EGFPs) (*Ishikawa et al., 2011*) showed that both IRE1α/β-DKO and XBP1-KO medaka exhibited three defects during early embryonic development, namely a short tail (white outline in *Figure 2C and D*, quantification in *Figure 3A*) which synthesizes and secretes a large amount of extracellular matrix proteins, failure of liver development (yellow arrowhead in *Figure 2C and D*) which synthesizes and secretes a large amount of proteins circulating in blood, and failure of hatching gland development (red arrowhead in *Figure 2C and D*) which synthesizes and secretes a large amount of hatching enzymes (proteases). Quantitative RT-PCR-mediated measurements revealed that the levels of Fabp10a mRNA encoding fatty-acid-binding protein 10a, a marker of liver (*Venkatachalam et al., 2009*), were indeed reduced to one-third in IRE1α/β-DKO embryos compared with IRE1β-single KO embryos, as well as in XBP1-KO embryos compared with WT embryos (*Figure 3B*).

Surprisingly, however, although the level of LCE mRNA encoding one of the two hatching enzymes (*Yasumasu et al., 1992*) was reduced to one-third in IRE1α/β-DKO embryos compared with IRE1β-single KO embryos, it was reduced to less than 10% in XBP1-KO embryos compared with WT embryos (*Figure 3B*), meaning that KO of XBP1, a transcription factor downstream of IRE1α/β, produces a much more severe phenotype than DKO of IRE1α/β in the hatching gland. This notion was fully supported by the markedly decreased fluorescence of EGFP, which is under the control of the LCE promoter (P$_{LCE}$-BiP), in the hatching gland of XBP1-KO medaka, compared with the mildly decreased fluorescence of EGFP in the hatching gland of IRE1α/β-DKO medaka (*Figure 3C, D and E*), and by the fact that XBP1-KO embryos could not hatch, whereas 70% of IRE1α/β-DKO embryos hatched (*Figure 3F*). Levels of other mRNAs were not significantly altered by the deletion of XBP1 (*Figure 3G*), namely Mcmlc2 mRNA encoding cardiac myosin light chain 2, a marker of heart (*Shimada et al., 2009*); Ins1 mRNA encoding insulin, a marker of pancreatic β-cells; Fabp2a mRNA encoding fatty acid binding protein 2a, a marker of intestine (*Parmar and Wright, 2013*); MyoD mRNA encoding one of the myogenic regulatory factors, a marker of myotome (*Liang et al., 2008*); and NeuN mRNA encoding a neuronal nuclear antigen, a marker of neuron (*Won et al., 2016*). We concluded that the IRE1-XBP1 branch is required for proper development of the three secretory organs (tail, liver and hatching gland).

## Markedly more severe phenotype in XBP1-KO medaka is not due to hyper-activation of IRE1-mediated signaling independent of XBP1

To explain phenotypic differences between IRE1α/β-DKO and XBP1-KO medaka we sought the possibility that the IRE1-mediated signaling independent of XBP1, namely the JNK and RIDD pathways (see *Figure 1A*), might be hyper-activated in the hatching gland of XBP1-KO medaka, leading to apoptosis (see Discussion for the reason why the effect of XBP1-KO is so profound in the hatching gland). To this end, we introduced IRE1α/β double deletion into XBP1-KO medaka to create IRE1α/IRE1β/XBP1-triple KO (TKO) medaka, and intended to clarify whether the removal of IRE1α/β from XBP1-KO medaka improves development of the three organs. Because we found that the deletion of IRE1α had more profound effect than that of IRE1β on physiological ER stress-induced splicing of XBP1 mRNA in embryos at 3 dpf (*Figure 4A*, compare lane 2 with lane 3), we first created IRE1β/XBP1-DKO medaka (*Figure 4Bc*), followed by the creation of IRE1α/IRE1β/XBP1-TKO medaka (*Figure 4Bd*). Analysis showed that there were no significant differences among XBP1-KO (data not shown in *Figure 4*), IRE1β/XBP1-DKO and IRE1α/IRE1β/XBP1-TKO in tail length (*Figure 4C*, compare lane 3 with lane 4), level of Fabp10a mRNA (*Figure 4D*, compare lane 3 with lane 4) and LCE

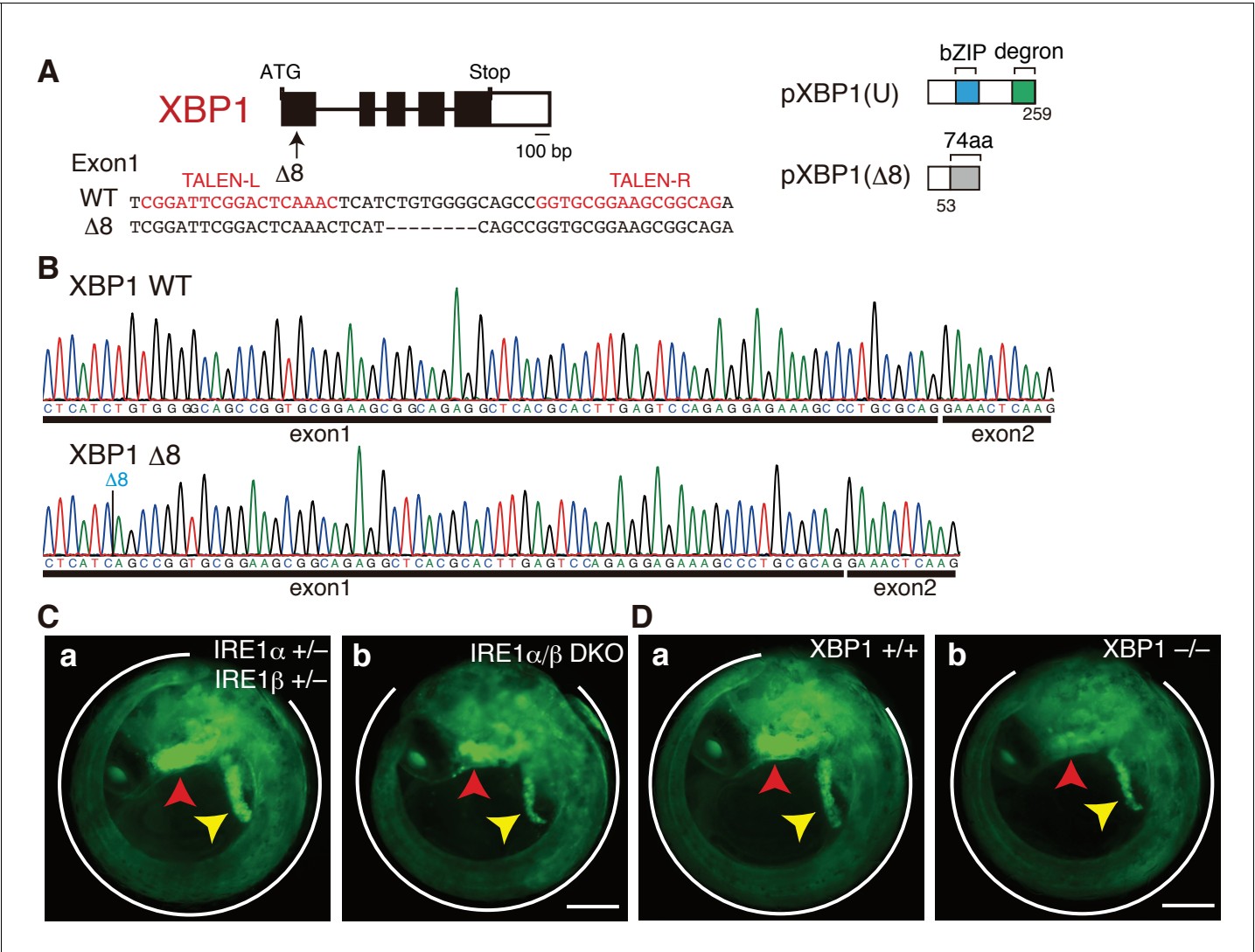

**Figure 2.** Construction of XBP1-KO medaka and fluorescence microscopic analysis of IRE1α/β-DKO and XBP1-KO medaka. (A) Schematic representation of the WT and Δ8-mutant XBP1 genes as well as their translational products pXBP1(U) and pXBP1(Δ8). bZIP denotes a basic leucine zipper, whereas the degron denotes a region which targets pXBP1(S) to proteasomal degradation. The grey box of 74 aa in pXBP1(Δ8) indicates foreign aa added to the C-terminus of the truncated XBP1 protein of 53 aa due to the frame shift at aa 54. (B) RT-PCR products corresponding to a part of XBP1 mRNA expressed in embryo at 5 dpf of WT (top) and XBP1-KO (bottom) medaka were sequenced. The position of expected deletion is shown in blue. Note that they do not have an intron. (C) Fluorescence microscopic analysis of IRE1α+/− IRE1β+/− and IRE1α/β-DKO embryos at 5 dpf which were obtained by crossing male IRE1α+/− IRE1β+/− medaka with female IRE1α+/− IRE1β-/- medaka, both carrying P<sub>BiP</sub>-EGFP. The white outline, red arrowhead and yellow arrowhead point to the tail, hatching gland and liver, respectively; this also applies to (D), 4B, 8A, 8B, 8C, 9B, and 12B. Scale bar: 250 μm. (D) Fluorescence microscopic analysis of XBP1+/+ and XBP1−/− embryos at 5 dpf obtained by incrossing male and female XBP1+/-medaka, both carrying P<sub>BiP</sub>-EGFP. Scale bar: 250 μm.

DOI: https://doi.org/10.7554/eLife.26845.003

mRNA (*Figure 4E*, compare lane 3 with lane 4) or hatching rates (*Figure 4F*, compare lane 3 with lane 4). Thus, hyper-activation of XBP1-independent signaling from IRE1 cannot explain the much more severe defect in the hatching gland of XBP1-KO medaka than in IRE1α/β-DKO medaka.

Instead, we developed a different scenario for the phenotypic difference between IRE1α/β-DKO and XBP1-KO medaka by determining the level of IRE1α mRNA encoding a major regulator of physiological ER stress-induced splicing of XBP1 mRNA (see *Figure 4A*), as well as the level of spliced XBP1 mRNA in embryos of various genotypes at 0.5, 0.7 and 1 dpf. The deletion of XBP1 had no effect on the level of IRE1α mRNA (*Figure 5A*, lanes 1, 2, 5, 6, 9, and 10), whereas the deletion of

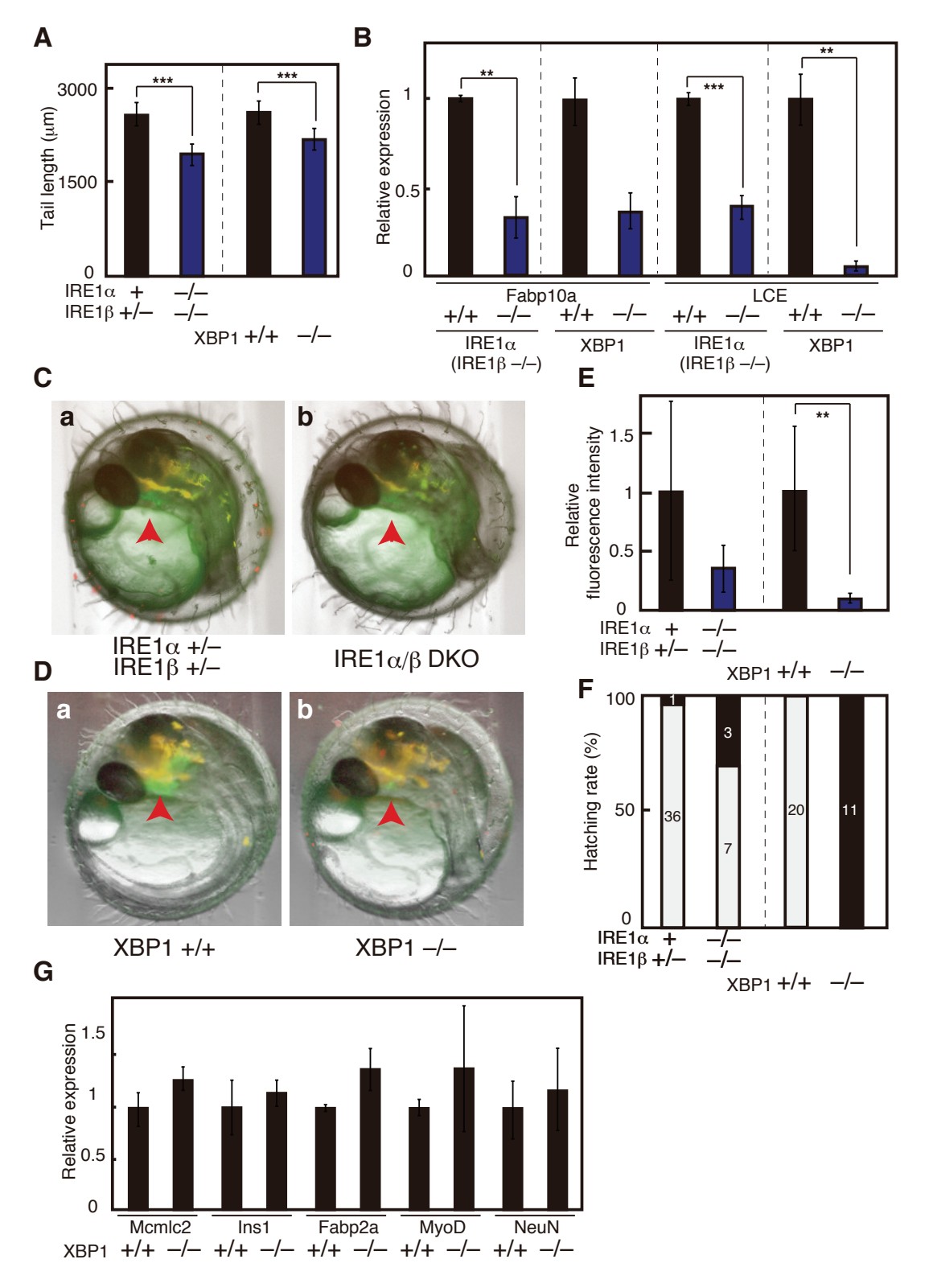

**Figure 3.** Effect of IRE1α/β-DKO and XBP1-KO on medaka development. (**A**) Tail lengths of the indicated genotypes shown in *Figure 2C and D* were measured (n > 3) and are expressed as means with SD (error bars). IRE1α + implies IRE1α+/+ or IRE1α+/-. *p<0.05, **p<0.01, ***p<0.001 for all figures. (**B**) Total RNA prepared from embryos at 5 dpf of the indicated genotypes shown in *Figure 2C and D* were subjected to quantitative RT-PCR to determine the levels of Fabp10a mRNA and LCE mRNA, which were normalized with the level of β-actin mRNA (n = 3). The mean value of normal fish is

*Figure 3 continued on next page*

*Figure 3 continued*
set as 1, and error bars indicate SD. (C) Fluorescence microscopic analysis of IRE1α+/− IRE1β+/− and IRE1α/β-DKO embryos at 3 dpf obtained by crossing male IRE1α+/− IRE1β+/− medaka with female IRE1α+/− IRE1β−/− medaka, both carrying P$_{LCE}$-EGFP. The red arrowhead indicates the hatching gland; this also applies to (D). (D) Fluorescence microscopic analysis of XBP1+/+ and XBP1−/− embryos at 3 dpf obtained by incrossing male and female XBP1+/- medaka, both carrying P$_{LCE}$-EGFP. (E) Fluorescence intensities in hatching glands shown in (C) and (D) were quantified. The mean value of normal fish is set as 1, and error bars indicate SD. (F) Hatching rates of embryos of the indicated genotypes were determined. Grey and black boxes indicate hatched and dead embryos, respectively, and the number in each box indicates actual data. (G) Levels of mRNA encoding Mcmlc2, Ins1, Fabp2a, MyoD and NeuN in embryos of the indicated genotypes were determined at 5 dpf and normalized with the level of β-actin mRNA. They are expressed as described in (B) (n = 3).
DOI: https://doi.org/10.7554/eLife.26845.004

IRE1α on the background of IRE1β-KO mildly reduced the level of IRE1α mRNA (*Figure 5A*, lanes 3, 4, 7, 8, 11, and 12), probably due to nonsense-mediated mRNA decay of IRE1α C156X mRNA (see *Figure 1B*). In contrast, the deletion of XBP1 markedly reduced the level of spliced XBP1 mRNA (*Figure 5B*, lanes 1, 2, 5 and 6; *Figure 5C*, lanes 1 and 2) as expected, probably due to nonsense-mediated mRNA decay of XBP1 Δ8 mRNA, which has the stop codon immediately after aa127 (see *Figure 2A*). Importantly, however, the deletion of IRE1α against the background of IRE1β-KO did not significantly affect the level of spliced XBP1 mRNA at 0.5 dpf, the early gastrula stage in medaka (*Figure 5C*, compare lane 3 with lane 4). We consider that maternal IRE1α WT mRNA derived from IRE1α+/− IRE1β−/− eggs was translated to produce IRE1α (protein), which initiated splicing of XBP1 mRNA transcribed from the XBP1 locus (zygotic XBP1 mRNA), in IRE1α−/− IRE1β−/− (XBP1 +/+) embryos at 0.5 dpf. The presence of spliced XBP1 mRNA (*Figure 5C*, lane 4) and thereby the presence of pXBP1(S) in embryos at 0.5 dpf stimulated development of the hatching gland at least partially in IRE1α/β-DKO medaka. In contrast, the hatching gland in XBP1-KO medaka developed very poorly due to the absence of spliced XBP1 mRNA (*Figure 5C*, lane 2) and thereby the absence of pXBP1(S) in embryos at 0.5 dpf (see Discussion for more detailed explanation and Figure 10C and D for unambiguous demonstration of this maternal effect).

## Constitutive expression of the spliced form of XBP1 fully rescues the defects observed in XBP1-KO medaka

Unspliced XBP1 mRNA is converted to spliced XBP1 mRNA by IRE1-mediated unconventional splicing in response to ER stress (*Figure 6B*). Activated IRE1 thus cleaves unspliced XBP1 mRNA at two characteristic stem-7-nucleotide-loop structures as shown in *Figure 6Ea*. Spliced XBP1 mRNA has lost such two stem-loop structures after the removal of the 26-nucleotide-intron as shown in *Figure 6Eb*.

We next aimed to determine whether or not the three defects observed in XBP1-KO medaka are rescued by constitutive expression of pXBP1(S), the spliced form of XBP1 (*Figure 6C*). We also intended to determine whether pXBP1(U), the unspliced form of XBP1, plays a role in the IRE1-mediated signaling, and whether removal of the entire exon 4, consisting of 167 nucleotides, by spliceosome-mediated alternative splicing (*Figure 6B*) plays an important role in XBP1 function, as has been proposed in gastrula- and early neurula-stage embryos of *Xenopus laevis* (*Cao et al., 2006*).

To this end, we introduced three deletions (Δ15, Δ26 and Δ4) into exon 4 containing IRE1-mediated splice sites in the XBP1 locus (*Figure 6A*) utilizing the TALEN and CRISPR-Cas9 methods (*Figure 6D*). WT and resulting mutant XBP1 mRNA was prepared from each embryo and converted to XBP1 cDNA, and respective XBP1 protein was N-terminally tagged with c-myc. Plasmid to express the respective XBP1 protein was transfected into the human colorectal carcinoma cell line HCT116 together with or without plasmid to express medaka IRE1α, and their cell lysates were analyzed by Immunoblotting (*Figure 7A and B*). Coexpression of medaka IRE1α markedly increased the level of medaka pXBP1(S) in the case of WT XBP1 cDNA, as expected (*Figure 7A*, compare lane 4 with lane 3). It should be noted that the level of medaka pXBP1(U) was very low and became visible only after long exposure (*Figure 7B*, compare lane 5 with lane 1).

The Δ15 mutant mRNA has lost the two stem-loop structures and that the Δ15 mutant protein lacked only five amino acids immediately C-terminal to the bZIP domain (*Figure 6Ec*). Thus, the Δ15 mutant protein designated pXBP1(U$^C$) represents constitutively expressed pXBP1(U) which is not further modified in response to ER stress (*Figure 6F*). Indeed, immunoblotting analysis revealed

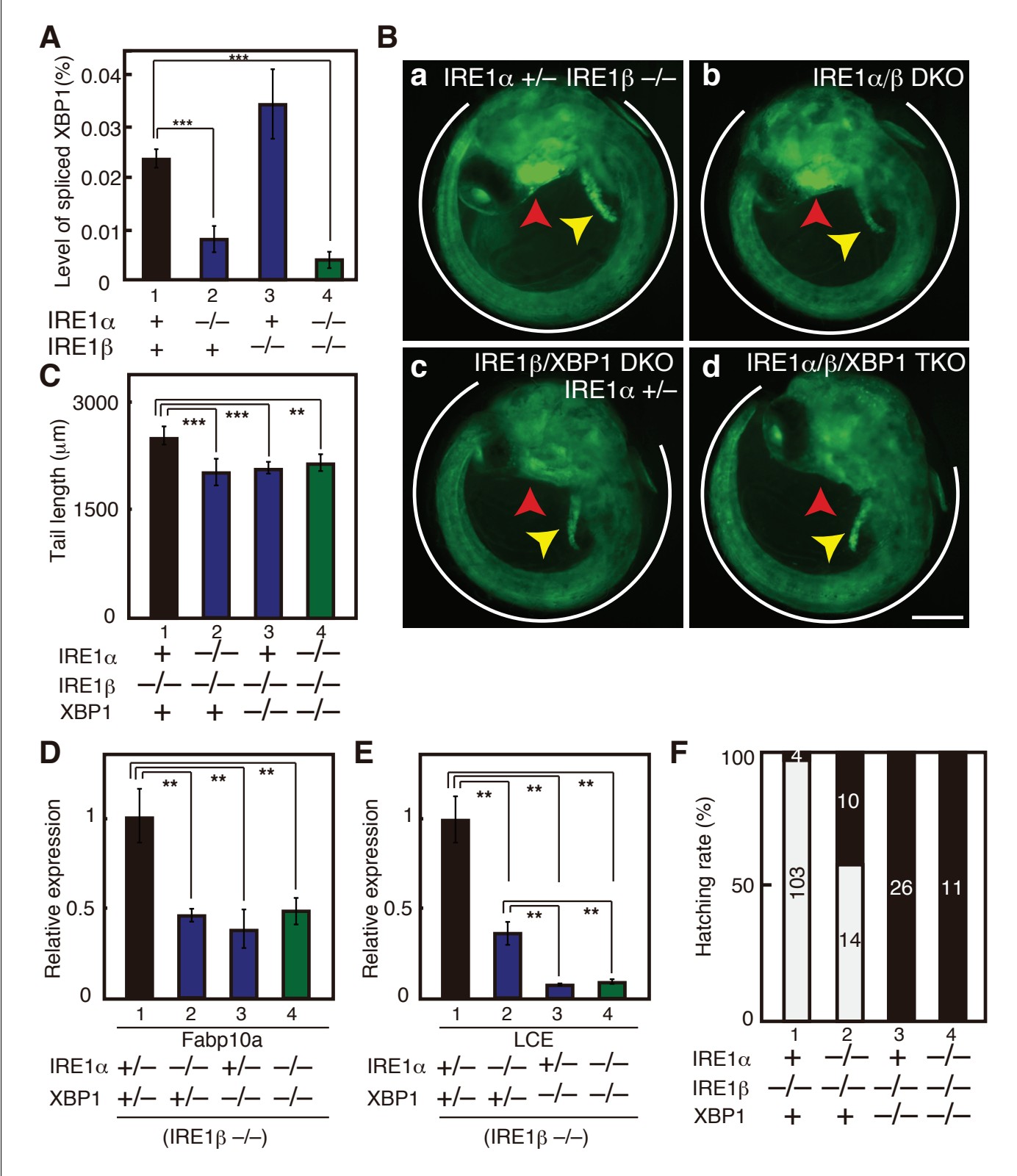

**Figure 4.** Effect on deleting IRE1α/β on phenotypes of XBP1-KO medaka. (**A**) Total RNA prepared from embryos of the indicated genotypes at 3 dpf were subjected to quantitative RT-PCR to determine absolute expression levels of spliced XBP1 mRNA and β-actin mRNA (n = 3). The level of spliced XBP1 mRNA relative to that of β-actin mRNA in the indicated genotypes is expressed as mean (%) with SD (error bars). (**B**) Fluorescence microscopic analysis of embryos at 5 dpf of the indicated genotypes which were obtained by incrossing male and female IRE1α+/− IRE1β−/− XBP1+/− medaka,

*Figure 4 continued on next page*

*Figure 4 continued*

both carrying P$_{BiP}$-EGFP. Scale bar: 250 µm. (C) Tail lengths of the indicated genotypes were measured and are expressed as in *Figure 3A* (n > 3). (D) Level of Fabp10a mRNA in embryos of the indicated genotypes at 5 dpf was determined, normalized with the level of β-actin mRNA and expressed as described in *Figure 3B* (n = 3). (E) Level of LCE mRNA in embryos of the indicated genotypes at 5 dpf was determined, normalized with the level of β-actin mRNA and expressed as described in *Figure 3B* (n = 3). (F) Hatching rates of embryos of the indicated genotypes were determined and are expressed as described in *Figure 3F*.

DOI: https://doi.org/10.7554/eLife.26845.005

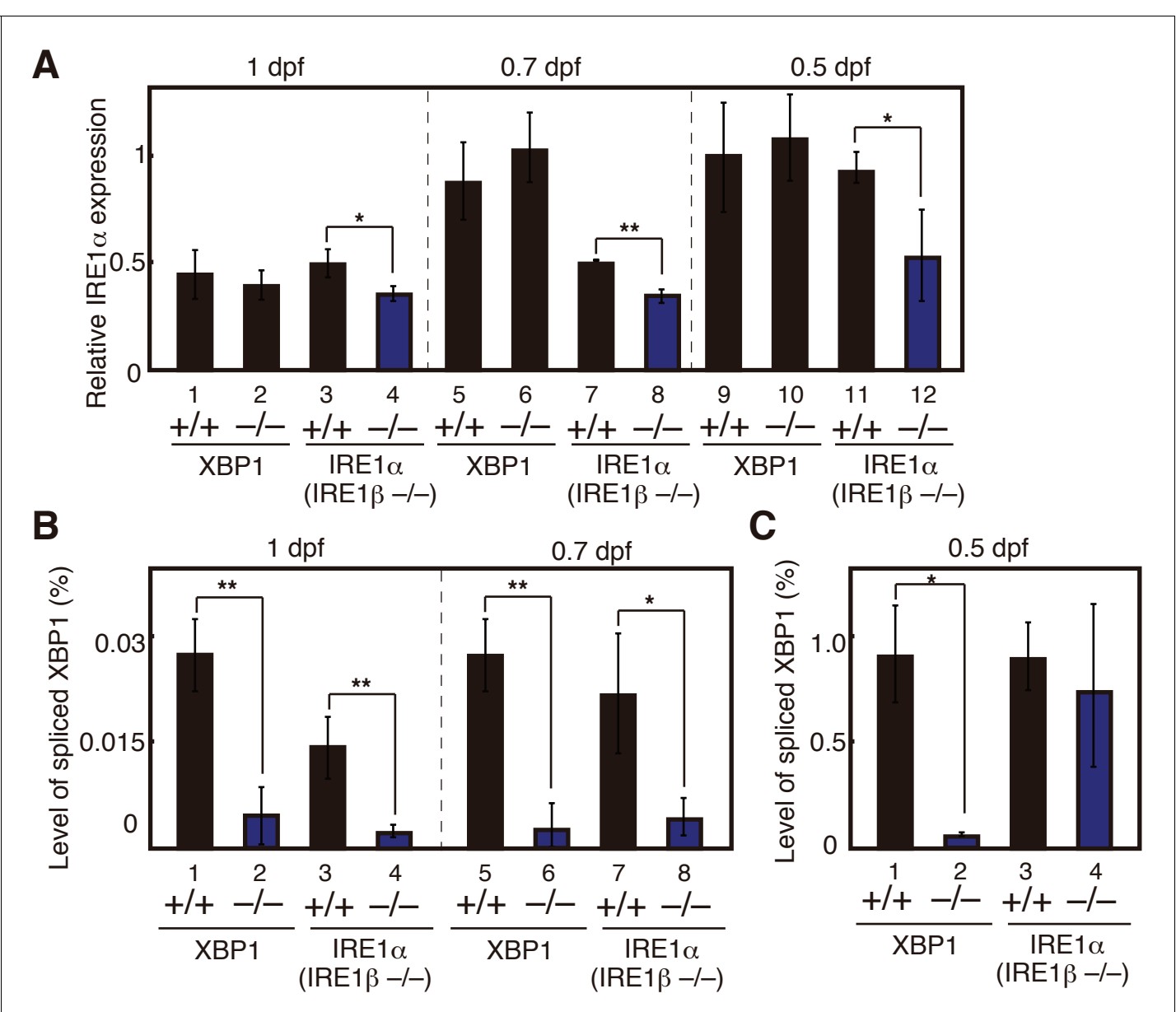

**Figure 5.** Maternal effect on expression levels of IRE1α mRNA and spliced XBP1 mRNA. (A) – (C) Embryos were obtained by incrossing male and female XBP1+/− medaka or by incrossing male and female IRE1α+/− IRE1β−/− medaka. (A) Level of IRE1α mRNA in embryos at indicated dpf of the indicated genotypes was determined and normalized with the level of β-actin mRNA as described in *Figure 3B* (n = 3). The mean value in XBP1+/+ embryos at 0.5 dpf (lane 9) is set as 1, and error bars indicate SD. (B) (C) Level of spliced XBP1 mRNA relative to that of β-actin mRNA at indicated dpf of the indicated genotypes was determined and expressed as described in *Figure 4A* (n = 3).

DOI: https://doi.org/10.7554/eLife.26845.006

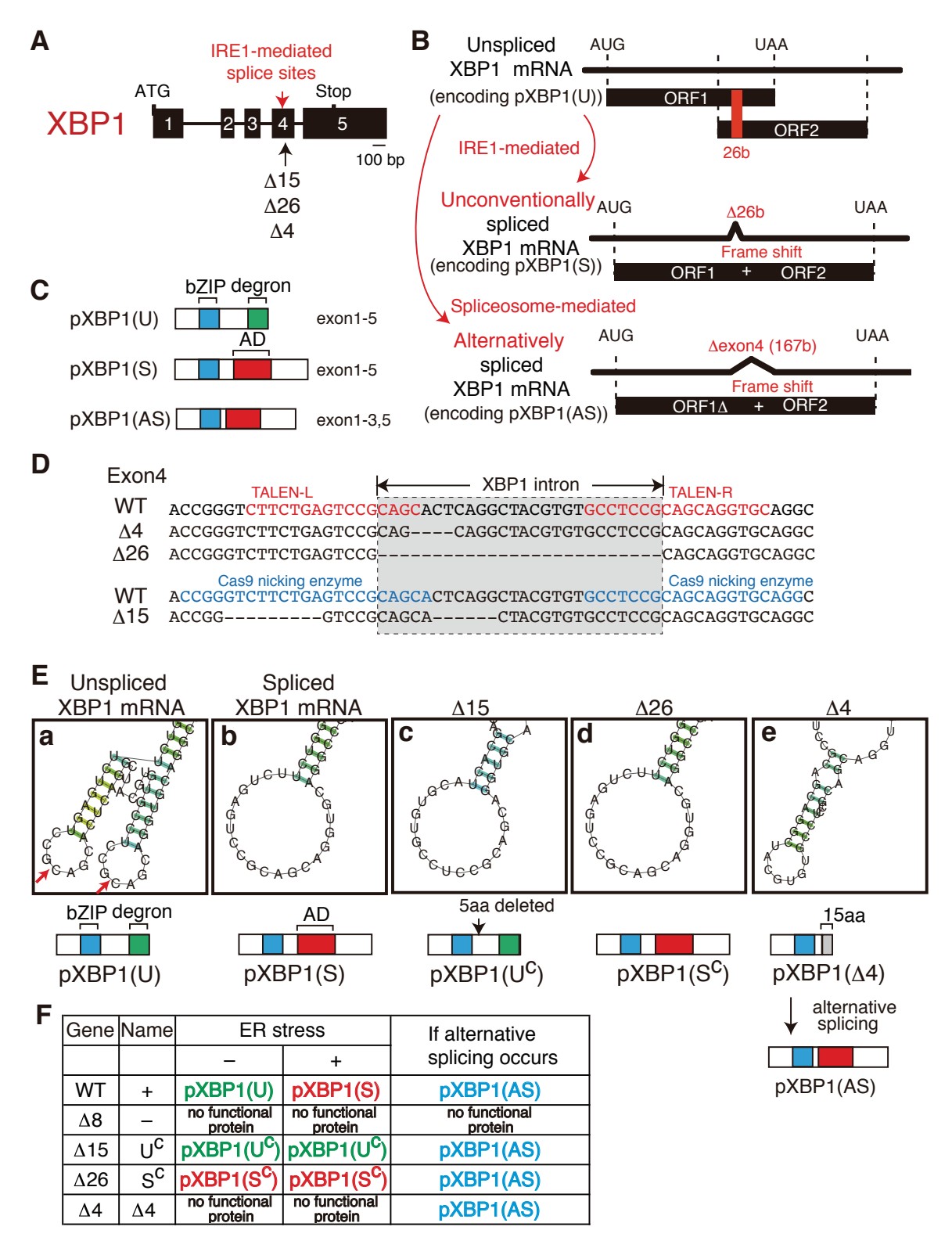

**Figure 6.** Construction of mutant alleles to express various forms of XBP1. (**A**) Schematic representation of the XBP1 gene as well as positions of the IRE1-mediated splice sites and three deletion mutants (Δ15, Δ26, Δ4). (**B**) Schematic representation of IRE1-mediated, unconventional splicing of XBP1 mRNA which removes the 26 b in exon 4 to produced pXBP1(S) as well as spliceosome-mediated alternative splicing of XBP1 mRNA which removes the entire exon 4 of 167 b to produce pXBP1(AS). pXBP1(U) is produced from unspliced XBP1 mRNA. (**C**) Schematic representation of pXBP1(U) (unspliced

*Figure 6 continued on next page*

*Figure 6 continued*

form), pXBP1(S) (spliced form) and pXBP1(AS) (alternatively spliced form). AD denotes transcriptional activation domain. (D) Sequences of a part of exon 4 in which three deletions were introduced by the TALEN method (Δ4 and Δ26) or CRISPR-Cas9 method (Δ15). The position of the XBP1 intron is also shown. (E) CentroidFold-predicted secondary structures of XBP1 mRNA around the IRE1-mediated splice sites (shown by two red arrows in a) produced by three deletion mutants (Δ15, Δ26, and Δ4) in comparison with those of unspliced and spliced XBP1 mRNA. The Δ15 and Δ26 deletion mutants produce constitutively expressed pXBP1(U$^C$) and pXBP1(S$^C$), respectively, which are not modified further after ER stress, as summarized in (F). The Δ4 deletion mutant produces pXBP1(Δ4) containing foreign 15 aa depicted as the grey box after aa165, which is inactive due to the absence of AD, but produces active pXBP1(AS) if spliceosome-mediated alternative splicing occurs, as summarized in (F). (F) Effects of ER stress and alternative splicing on expression of various forms of XBP1.

DOI: https://doi.org/10.7554/eLife.26845.007

constitutive expression of pXBP1(U$^C$) (*Figure 7B*, lane 7) and no production of pXBP1(S) even in the presence of medaka IRE1α (*Figure 7B*, lanes 4 and 8). The Δ26 mutant mRNA has lost 26 nucleotides which are identical to the XBP1 intron (*Figure 6D*). Thus, the Δ26 mutant protein designated pXBP1(S$^C$) represents constitutively expressed pXBP1(S) which is not further modified in response to ER stress (*Figure 6Ed and F*). This notion was firmly supported by immunoblotting analysis (*Figure 7A*, lanes 9 and 10). The Δ4 mutant mRNA has lost the two stem-loop structures and the Δ4 mutant protein is inactive due to the absence of AD, even if constitutively expressed (*Figure 7A*, lane 11), but is switched to an active protein designated pXBP1(AS) only if removal of the entire exon 4 by alternative splicing occurs (*Figure 6Ee and F*). Medaka IRE1α might have induced degradation of Δ15 or Δ4 mutant XBP1 mRNA, because the level of pXBP1(U$^C$) or pXBP1(Δ4) was decreased (*Figure 7A*, lane 12 and *Figure 7B*, lane 8). It should be noted that the Δ8 mutant protein (see *Figure 2A*) also behaved as expected in this immunoblotting analysis (*Figure 7A*, lanes 5 and 6).

We confirmed the presence of the expected deletion in respective XBP1 mRNA expressed in embryo at 5 dpf by direct sequencing of RT-PCR products (*Figure 7C*), namely Δ4, Δ15, and Δ26 in XBP1 cDNA prepared from Δ4/Δ4 embryo (see *Figure 8A*), from Δ15/Δ8 (U$^c$/-) embryo (see *Figure 8B*; note that XBP1 Δ8 mRNA is unstable, probably due to nonsense-mediated mRNA decay), and from S$^C$/S$^C$ embryo (see Figure 12Ac), respectively. We found that embryos, in which both alleles of the XBP1 locus contained the Δ4 mutation (*Figure 8Ab*), exhibited three organ defects, namely short tail (*Figure 8Ea*), developmental failure of the liver and hatching gland (*Figure 8Fa*), as well as inability to hatch (*Figure 8Ga*), similarly to XBP1-KO (XBP1Δ8/Δ8) embryos (*Figures 2D*, *3A, B and F*), indicating that alternative splicing does not take place during embryonic development in medaka. We then obtained embryos in which one allele of the XBP1 locus has a Δ15 mutation (XBP1U$^C$) and the other allele has the Δ8 mutation (XBP1-), by which the level of spliced XBP1 mRNA was markedly decreased (*Figure 8Da*). We found that these XBP1U$^C$/- embryos exhibited three organ defects (*Figure 8Bb*), namely short tail (8Eb), developmental failure of the liver and hatching gland (*Figure 8Fb*), as well as inability to hatch (*Figure 8Gb*), similarly to XBP1-KO embryos (*Figures 2D*, *3A, B and F*), indicating that pXBP1(U) plays little role during embryonic development in medaka. We further obtained embryos in which one allele of the XBP1 locus had a Δ26 mutation (XBP1S$^C$) and the other allele had the Δ8 mutation (XBP1-), by which the level of spliced XBP1 mRNA was markedly increased (*Figure 8Db*). Importantly, the three organ defects (*Figure 8Cb*) were completely rescued in XBP1S$^C$/- embryos (*Figure 8Ec,Fc*) and all XBP1S$^C$/- embryos hatched (*Figure 8Gc*). These results clearly show that XBP1 mRNA must be spliced for downstream signaling.

## Constitutive expression of the spliced form of XBP1 fully rescues the defects observed in IRE1α/β-DKO medaka

We further intended to determine whether or not the three defects observed in IRE1α/β-DKO medaka are rescued by constitutive expression of pXBP1(S), similarly to the case of XBP1-KO medaka. Because IRE1α/β-DKO medaka die within 2 weeks after fertilization (see Figure 10A) and therefore cannot be used for mating, we determined whether IRE1α/β-DKO medaka become fertile if the Δ26 mutation (XBP1S$^C$) is introduced into one allele of the XBP1 locus. To this end, male IRE1α+/− IRE1β−/− (XBP1+/+) were crossed with female (IRE1α+/+ IRE1β+/+) XBP1S$^C$/+ medaka, resulting in the production of male and female IRE1α+/− IRE1β+/− XBP1S$^C$/+ medaka Then, male

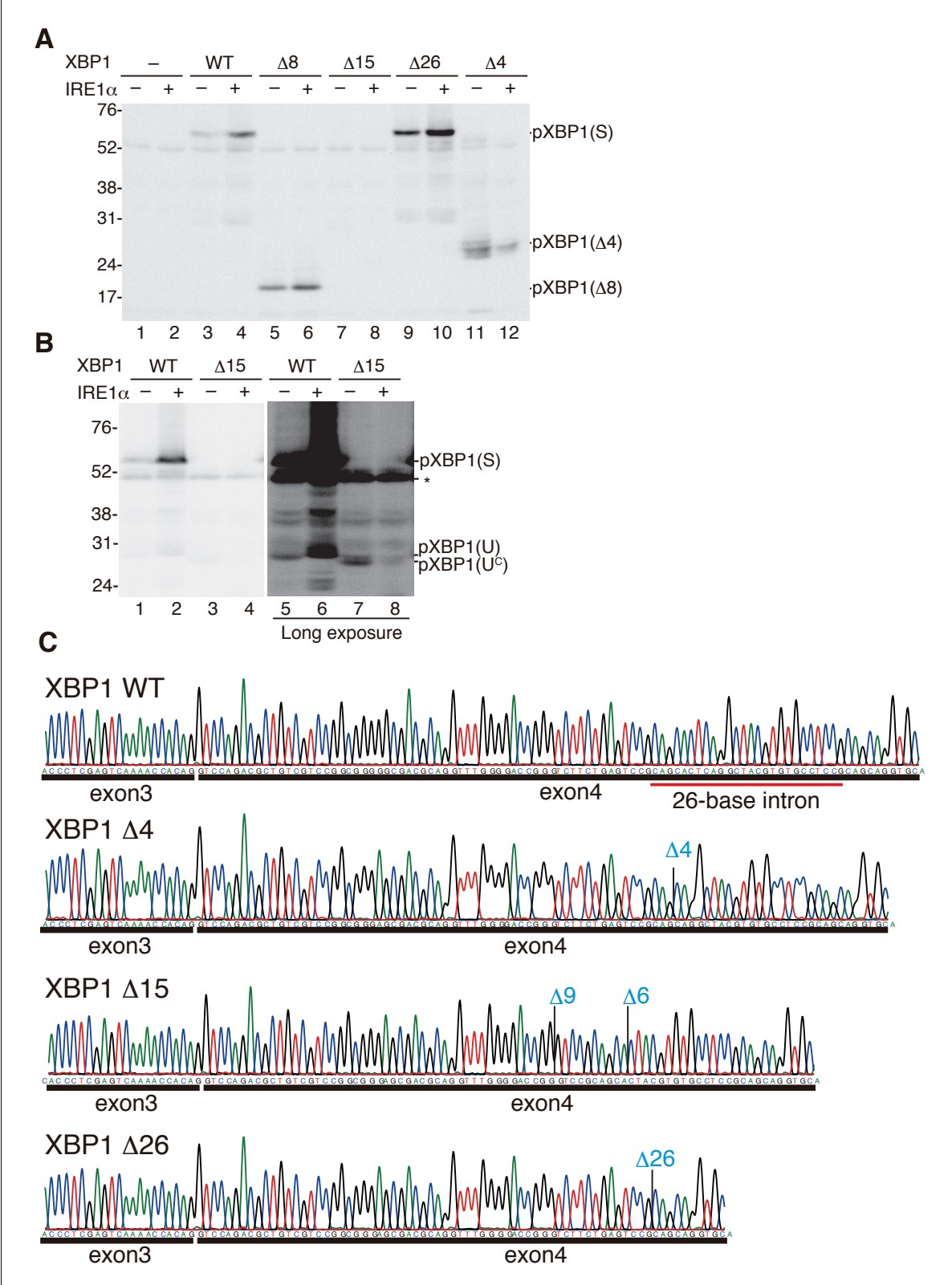

**Figure 7.** Characterization of various forms of XBP1. (**A**) HCT116 cells were transfected with pCMV-myc(-), pCMV-myc-medakaXBP1(WT), pCMV-myc-medakaXBP1(Δ8), pCMV-myc-medakaXBP1(Δ15), pCMV-myc-medakaXBP1(Δ26) or pCMV-myc-medakaXBP1 (Δ4) together with (+) or without (-) pcDNA-medakaIRE1α. MG132 was added for 4 hr prior to harvest. 30 hr after transfection, cell lysates were prepared and analyzed by immunoblotting using anti-myc antibody. (**B**) HCT116 cells were transfected with pCMV-myc-medakaXBP1(WT), pCMV-myc-medakaXBP1(Δ15) together with (+) or without (-)

*Figure 7 continued on next page*

Figure 7 continued
pcDNA-medakaIRE1α. MG132 was added for 2 hr prior to harvest. 24 hr after transfection, cell lysates were prepared and analyzed by immunoblotting using anti-myc antibody. (C) RT-PCR products corresponding to a part of XBP1 mRNA expressed in respective embryo at 5 dpf of WT, Δ4/Δ4, Δ15/Δ8, and Δ26/Δ26 XBP1 medaka were sequenced. The positions of expected deletions are shown in blue. Note that they do not have an intron.
DOI: https://doi.org/10.7554/eLife.26845.008

IRE1α+/− IRE1β-/- XBP1+/+ medaka were crossed with female IRE1α+/− IRE1β+/− XBP1S$^C$/+ medaka, resulting in the production of one female IRE1α-/- IRE1β-/- XBP1S$^C$/+ medaka among 48 fishes which survived until 2 months after hatching (*Figure 9A*, shown in red).

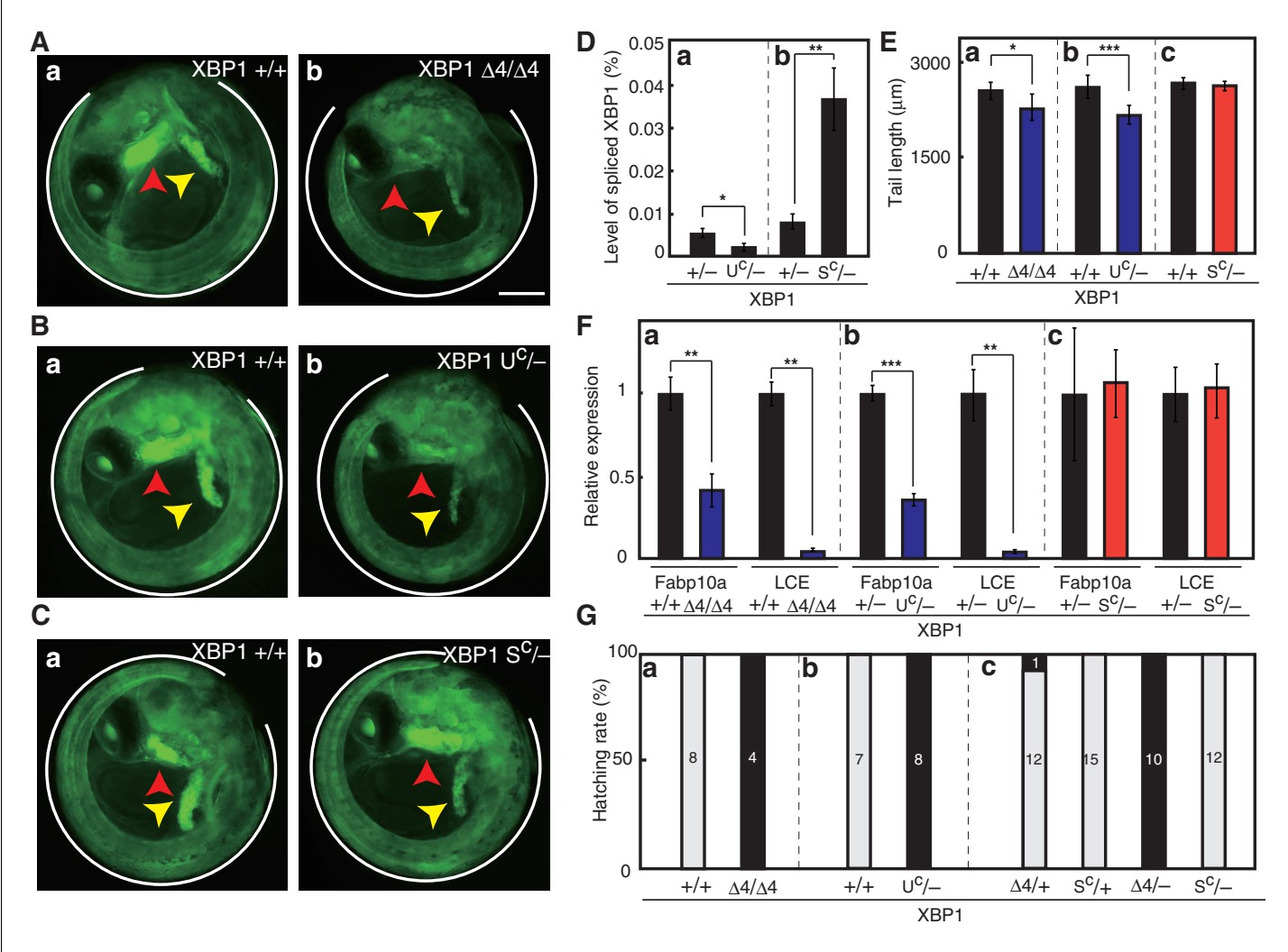

**Figure 8.** Effect of expression of various forms of XBP1 on medaka development. (A) Fluorescence microscopic analysis of XBP1+/+ and XBP1Δ4/Δ4 embryos at 5 dpf which were obtained by incrossing male and female XBP1+/Δ4 medaka, both carrying P$_{BiP}$-EGFP. Scale bar: 250 µm. (B) Fluorescence microscopic analysis of XBP1+/+ and XBP1U$^C$/- embryos at 5 dpf which were obtained by crossing male XBP1U$^C$/+and female XBP1+/- medaka, both carrying P$_{BiP}$-EGFP. (C) Fluorescence microscopic analysis of XBP1+/+ and XBP1S$^C$/- embryos at 5 dpf which were obtained by crossing male XBP1S$^C$/+ and female XBP1+/− medaka, both carrying P$_{BiP}$-EGFP. (D) Level of spliced XBP1 mRNA relative to that of β-actin mRNA in embryos of the indicated genotypes at 5 dpf was determined and is expressed as described in *Figure 4A* (n = 3). (E) Tail lengths of the indicated genotypes were measured and are expressed as in *Figure 3A* (n > 3). (F) Levels of Fabp10a mRNA and LCE mRNA in embryos of the indicated genotypes at 5 dpf were determined and normalized with the level of β-actin mRNA, and expressed as described in *Figure 3B* (n = 3). (G) Hatching rates of embryos of the indicated genotypes were determined and are expressed as described in *Figure 3F*.
DOI: https://doi.org/10.7554/eLife.26845.009

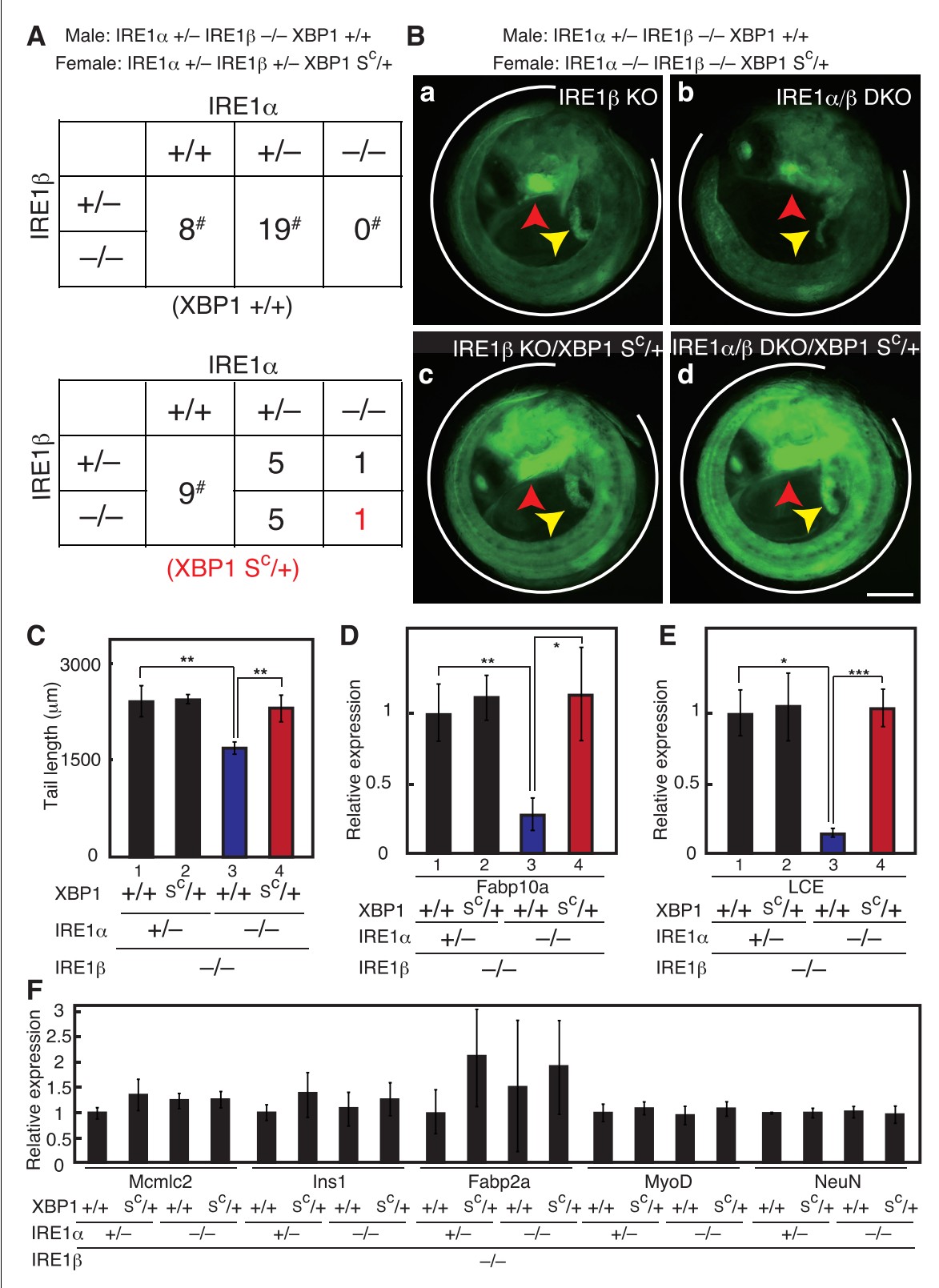

**Figure 9.** Complete rescue of IRE1α/β-DKO medaka by constitutive expression of pXBP1(S). (**A**) Male IRE1α+/− IRE1β−/− XBP1+/+ medaka were crossed with female IRE1α+/− IRE1β+/− XBP1S$^C$/+and the 48 resulting fishes were genotyped 2 months after hatching. The genotype of IRE1β was not determined for fishes marked with #. (**B**) Fluorescence microscopic analysis of embryos of the indicated genotypes at 5 dpf which were obtained by crossing male IRE1α+/− IRE1β−/− XBP1+/+ medaka with female IRE1α−/− IRE1β−/− XBP1S$^C$/+ medaka, both carrying P$_{BiP}$-EGFP. Scale bar: 250 μm.
*Figure 9 continued on next page*

*Figure 9 continued*

(C) Tail lengths of the indicated genotypes were measured and are expressed as in *Figure 3A* (n > 3). (D) Level of Fabp10a mRNA in embryos of the indicated genotypes at 5 dpf was determined, normalized with that of β-actin mRNA and is expressed as described in *Figure 3B* (n = 3). (E) Level of LCE mRNA in embryos of the indicated genotypes at 5 dpf was determined, normalized with that of β-actin mRNA and is expressed as described in *Figure 3B* (n = 3). (F) Levels of mRNA encoding Mcmlc2, Ins1, Fabp2a, MyoD and NeuN in embryos at 5 dpf of the indicated genotypes were determined and normalized with that of β-actin mRNA, and are expressed as described in *Figure 3B* (n = 3).

DOI: https://doi.org/10.7554/eLife.26845.010

These female IRE1α-/- IRE1β-/- XBP1S$^C$/+ medaka were then crossed with male IRE1α+/− IRE1β-/- XBP1+/+ medaka, resulting in the production of embryos with four different genotypes, as shown in *Figure 9B*. As shown previously (*Figure 2C*), IRE1α/β-DKO (XBP1+/+) embryos showed three organ defects at 5 dpf, namely short tail (*Figure 9C*, lane 3) as well as developmental failure of the liver (*Figure 9D*, lane 3) and hatching glad (*Figure 9E*, lane 3). Very importantly, introduction of the Δ26 mutation (XBP1S$^C$) into one allele of the XBP1 locus (XBP1S$^C$/+) in IRE1α/β-DKO medaka fully rescued these three defects (*Figure 9Bd and C*, lane 4; *Figure 9D*, lane 4; and *Figure 9E*, lane 4), although expression levels of various tissue-specific markers were not altered (*Figure 9F*). We therefore concluded that the main task of IRE1 is to produce the spliced form of XBP1 during growth and development of medaka.

Subsequent crossing confirmed that IRE1α/β-DKO XBP1S$^C$/+ medaka hatched and were alive for 2 months, unlike IRE1α/β-DKO XBP1+/+ medaka (*Figure 10A and B*). Importantly, fertility of both male and female IRE1α/β-DKO XBP1S$^C$/+ medaka allowed us to determine hatching rates of IRE1α/β-DKO XBP1+/+ embryos (see *Figure 3F*) after two types of mating. When male IRE1α/β-DKO XBP1S$^C$/+ medaka were crossed with female IRE1α+/− IRE1β−/− XBP1+/+ medaka, 30% of 10 IRE1α/β-DKO XBP1+/+ embryos containing maternal IRE1α hatched (*Figure 10C*, lane 3). In marked contrast, when male IRE1α+/− IRE1β-/- XBP1+/+ medaka were crossed with female IRE1α/β-DKO XBP1S$^C$/+ medaka, none of 11 IRE1α/β-DKO XBP1+/+ embryos not containing maternal IRE1α hatched (*Figure 10D*, lane 3). These results provide clear evidence for the maternal effect of IRE1α on development of the hatching gland.

We further conducted RNA-seq analysis of transcripts expressed in embryos of various genotypes at 5 dpf resulting from crossing male IRE1α−/− IRE1β−/− XBP1S$^C$/+ medaka with female IRE1α+/− IRE1β-/- XBP1+/+ medaka (*Figure 11A*), as described in Materials and methods. The entire data set obtained is deposited in the DNA Data Bank of JAPAN (http://www.ddbj.nig.ac.jp) with the accession number DRA006141. Among 19,979 medaka genes deposited in the ensemble database 11,979 genes have human counterparts and have been assigned gene ontology (GO). When the gene expression profile in embryos of IRE1β-KO (IRE1α+/− IRE1β−/− XBP1+/+) was compared with that in embryos of IRE1α/β-DKO (IRE1α−/− IRE1β-/- XBP1+/+), 200 and 26 genes were significantly up-regulated and down-regulated, respectively, in IRE1α/β-DKO (*Figure 11A*, left panel), with the q value of less than 0.05 (equivalent to p<0.000771). Importantly, when the gene expression profile in embryos of IRE1β-KO (IRE1α+/− IRE1β-/- XBP1+/+) was compared with that in embryos of IRE1α/β-DKO XBP1S$^C$/+ (IRE1α−/− IRE1β−/− XBP1S$^C$/+), 225 of 226 genes exhibited no significant differences (*Figure 11A*, right panel). The only exception was Fabp2a mRNA. However, we could not confirm the significance of this gene by quantitative RT-PCR (*Figure 9F*). We concluded that the observed full rescue of the defects in IRE1α/β-DKO by introduction of XBP1S$^C$ (*Figures 9B, C, D and E*) is ascribable to transcriptional activity of XBP1S$^C$.

Many genes among the 200 significantly up-regulated genes have functions related to the ER (*Figure 11B*), suggesting that embryos of IRE1α/β-DKO were ER stressed, and that such ER stress evoked in response to the complete absence of the IRE1 pathway was completely ameliorated by the introduction of XBP1S$^C$. Roles of 26 significantly down-regulated genes in the development of phenotypes of IRE1α/β-DKO medaka will be investigated and reported elsewhere.

## Hyper-activation of XBP1 mRNA splicing is detrimental to growth and development of medaka

We finally obtained embryos in which both alleles of the XBP1 locus contained the Δ26 mutation (XBP1S$^C$) and found that EGFP expression from the ER chaperone BiP promoter was highly enhanced by constitutive expression of pXBP1(S$^C$) from both alleles (*Figure 12Ac*). These XBP1S$^C$/

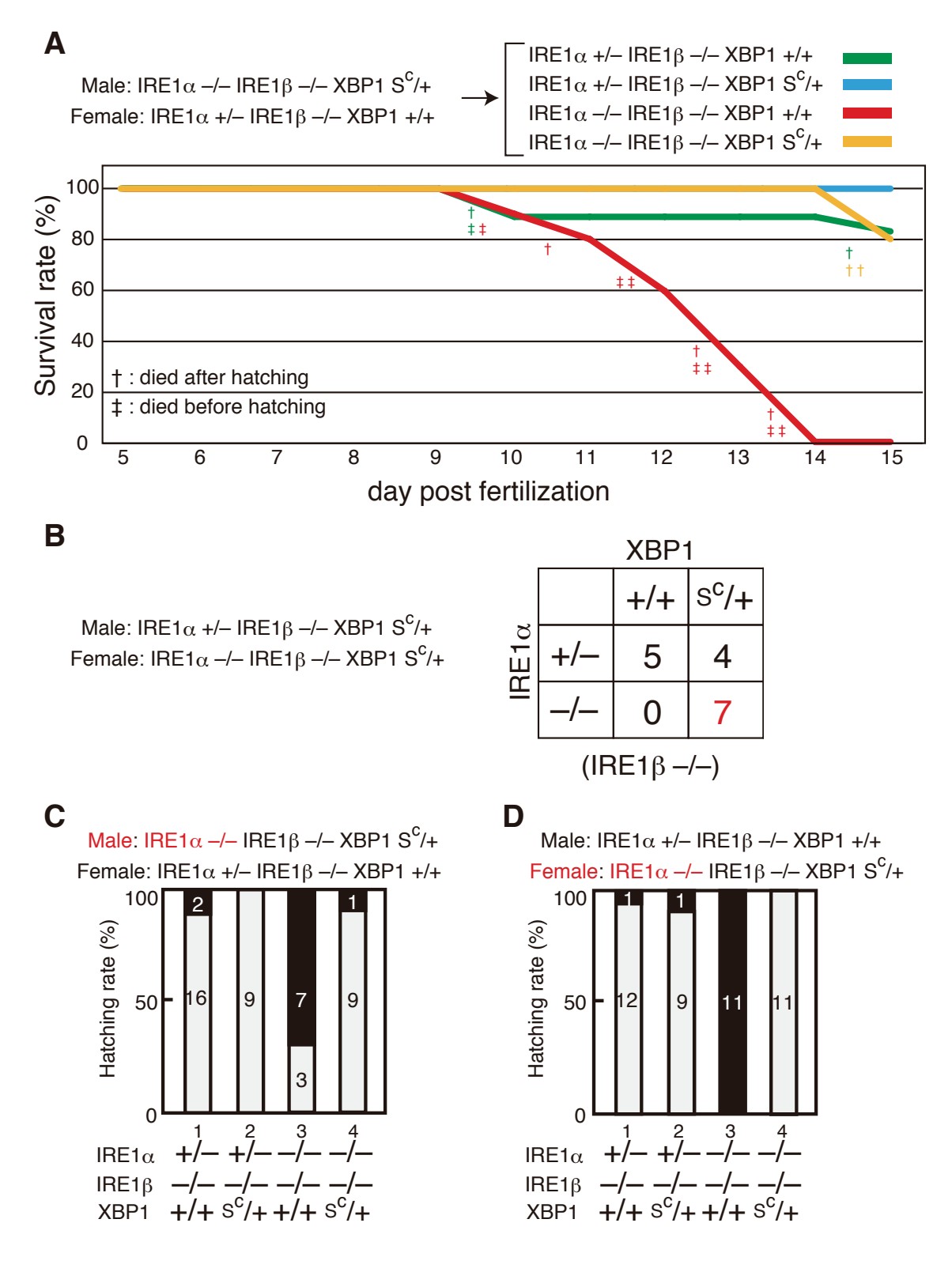

**Figure 10.** Survival rate and hatching rate of IRE1α/β-DKO medaka with or without the XBP1S$^c$ allele. (**A**) Male IRE1α-/- IRE1β-/- XBP1S$^C$/+ medaka were crossed with female IRE1α+/− IRE1β-/- XBP1+/+ medaka, and the resulting 47 embryos were grown. Dead fish and fish surviving until 15 dpf were genotyped. (**B**) Male IRE1α+/− IRE1β-/- XBP1S$^C$/+ medaka were crossed with female IRE1α-/- IRE1β-/- XBP1S$^C$/+ medaka and the 16 resulting fishes

*Figure 10 continued on next page*

*Figure 10 continued*
were genotyped 2 months after hatching. (C, D) Hatching rates of embryos of the indicated genotypes resulting from the indicated crossing were determined and are expressed as described in *Figure 3F*.
DOI: https://doi.org/10.7554/eLife.26845.011

S$^C$ embryos showed a slightly short tail (*Figure 12B and C*) but hatched (*Figure 12E*) even though the level of LCE mRNA was significantly decreased (*Figure 12D*). However, unlike XBP1S$^C$/+or XBP1S$^C$/- medaka (*Figure 12F*), XBP1S$^C$/S$^C$ medaka could not be found 2 months after hatching (*Figure 12G*), implying that hyper-activation of XBP1 mRNA splicing and the resulting excessive expression of pXBP1(S) is detrimental to the normal growth and development of medaka. XBP1 mRNA splicing must be under the control of IRE1, whose activation status is adjusted according to the strength and duration of ER stress.

## Discussion

Here, we constructed and analyzed IRE1α/β-DKO and XBP1-KO medaka and found that both showed defects in the development of three organs which synthesize and secrete a large amount of proteins, such as extracellular matrix proteins from tail, proteins circulating in blood from liver, and hatching enzymes from hatching gland. This confirms the critical importance of the IRE1-XBP1 branch of the UPR in maintenance of the protein quality control system operating in the ER, particularly in secretory organs, similar to the case in mice; proteins circulating in blood plus proteins for hematopoiesis from liver (*Reimold et al., 2000*), immunoglobulins from plasma cells (*Reimold et al., 2001*), digestive enzymes from exocrine pancreas (*Lee et al., 2005*), and lysozyme from intestinal epithelial cells (*Kaser et al., 2008*).

It is worth noting that only the defect in the hatching gland of XBP1-KO medaka was substantially more severe than that of IRE1α/β-DKO medaka among the three organs, even though XBP1 is a transcription factor downstream of IRE1α/β (*Figure 3*). We obtained XBP1+/+ and XBP1-/- embryos by incrossing male and female XBP1+/− medaka, in whose eggs WT mRNA and Δ8 mRNA were expressed. We consider that maternal XBP1 (both WT and Δ8) mRNA is degraded in eggs prior to fertilization because of the absence of spliced XBP1 mRNA in embryos at 0.5 dpf of XBP1-/- medaka (*Figure 5C*, lane 2), and consider that the spliced XBP1 mRNA detected in embryos at 0.5 dpf of XBP1+/+ medaka is derived from IRE1α-mediated splicing of zygotic XBP1 WT mRNA (*Figure 5C*, lane 1). The absence of pXBP1(S) in embryos at 0.5 dpf likely explain the very poor development of the hatching gland in XBP1-/- medaka (*Figure 3*), although LCE, HCE1 and HCE2 mRNAs encoding hatching enzymes do not appear to be direct targets of pXBP1(S) because they were not up-regulated in IRE1β-KO embryos by the introduction of XBP1S$^C$ (data not shown in *Figure 11*). In this connection, it is known that Nodals, members of the TGFβ family, are required for induction of mesodermal and endodermal progenitors during vertebrate gastrulation (*Feldman et al., 1998*; *Schier, 2003*) and for subsequent derivation of the notochord and hatching gland from the dorsal mesoderm in zebrafish (*Hagos and Dougan, 2007*). XBP1 is a direct target of Nodal signaling and is required for terminal differentiation of the hatching gland in zebrafish (*Bennett et al., 2007*). Thus, the presence of pXBP1(S) during vertebrate gastrulation (0.5–0.7 dpf in medaka) may be key to the development of the hatching gland. We speculate that ER stress which might be experienced in eggs would not be inherited into fertilized eggs as a form of spliced XBP1 mRNA whose translational product, pXBP1(S), may interfere developmentally regulated Nodal signaling.

We obtained IRE1α/β-DKO embryos by crossing male IRE1α+/- IRE1β+/- medaka with female IRE1α+/- IRE1β-/- medaka, in whose eggs IRE1α WT and C156X mRNA was expressed. We consider that maternal IRE1α (both WT and C156X) mRNA is transmitted to fertilized eggs because of the presence of spliced XBP1 mRNA in embryos at 0.5 dpf of both IRE1β-KO and IRE1α/β-DKO medaka (*Figure 5C*, lanes 3 and 4). Thus, the level of IRE1α mRNA detected in embryos at 0.5 dpf of IRE1β-KO medaka (*Figure 5A*, lane 11) would be the summation of maternal IRE1α (WT and C156X) mRNA and zygotic IRE1α WT mRNA, whereas the level of IRE1α mRNA detected in embryos at 0.5 dpf of IRE1α/β-DKO medaka (*Figure 5A*, lane 12) would be summation of maternal IRE1α (WT and C156X) mRNA and zygotic IRE1α C156X mRNA. Because the level of IRE1α mRNA in embryos at 0.5 dpf of IRE1α/β-DKO medaka was significantly lower than that in embryos at 0.5 dpf of IRE1β-KO

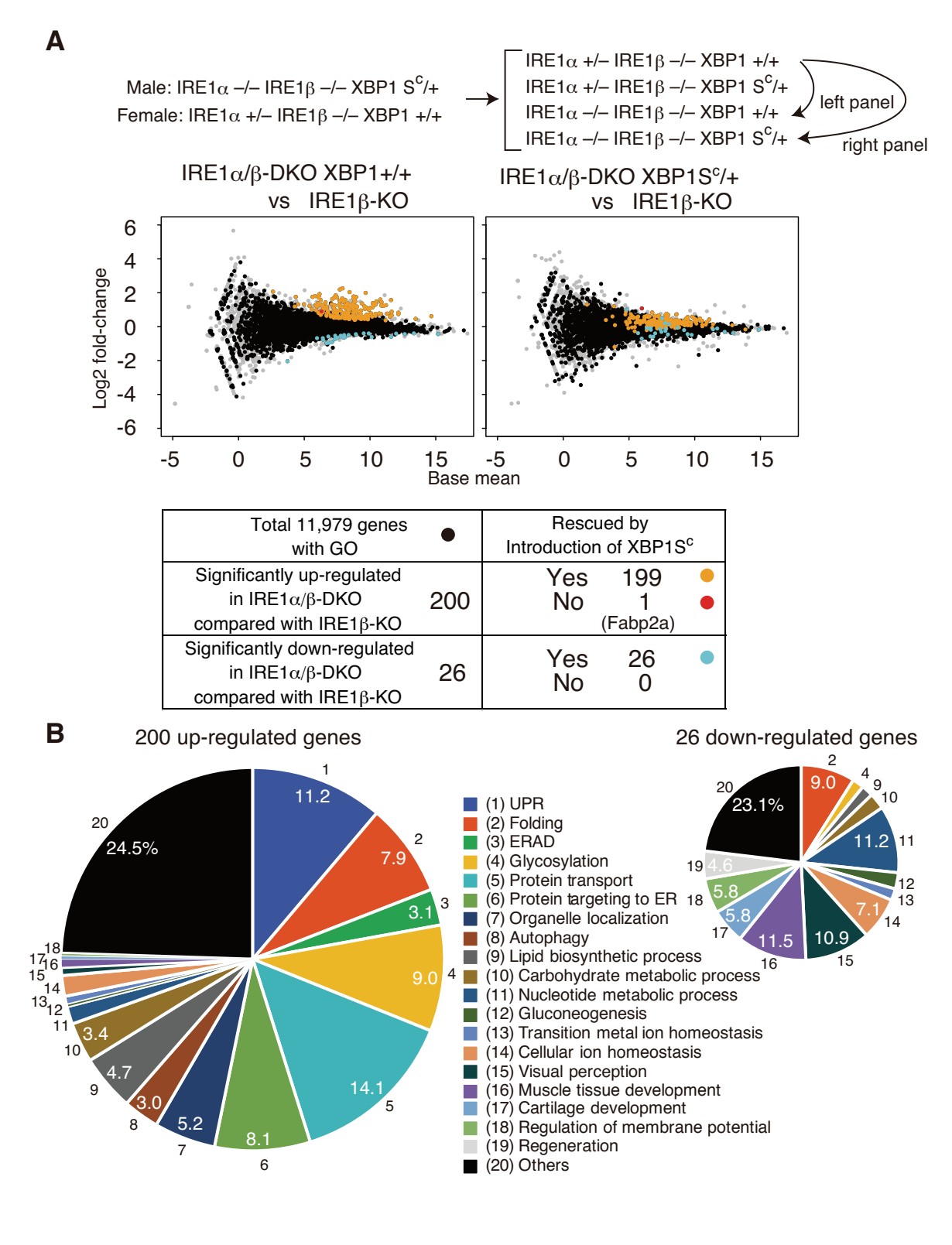

**Figure 11.** Transcriptomic comparison of IRE1β-KO, IRE1α/β-DKO XBP1+/+ and IRE1α/β-DKO XBP1S$^C$/+ embryos. (**A**) RNA prepared from embryos at 5 dpf of the indicated genotypes were subjected to RNA-seq analysis and the data obtained are expressed as MA plot. Genes whose expression was significantly altered in IRE1α/β-DKO compared with IRE1β-KO are colored and summarized in the table below. Grey dots denote genes without GO. n = 4 for IRE1β-KO and n = 3 for IRE1α/β-DKO XBP1+/+ and IRE1α/β-DKO XBP1S$^C$/+. (**B**) 200 significantly up-regulated and 26 significantly down-

**Figure 11 continued**

regulated genes are categorized based on their assigned GO. GO:0006986, GO:0030968, GO:0034620, GO:0034966, GO:0034976 and GO:0035967 for UPR (1); GO:0006457, GO:0016485 and GO:0051604 for Folding (2); GO:0030433, GO:0030970 and GO:0036503 for ERAD (3); GO:0006486, GO:0006487, GO:0009100, GO:0009101, GO:0018279, GO:0043413 and GO:0070085 for Glycosylation (4); GO:0006888, GO:0006890, GO:0006900, GO:0006901, GO:0006903, GO:0048193, GO:0048199, GO:0048207, GO:0048208 and GO:0090114 for Protein transport (5); GO:0006612, GO:0006613, GO:0006614, GO:0045047, GO:0070972, GO:0072594, GO:0072599, GO:0072657 and GO:0090150 for Protein targeting to ER (6); GO:0051640 and GO:0051656 for Organelle localization (7); GO:0006914 for Autophagy (8); GO:0006644, GO:0008610 and GO:0046486 for Lipid biosynthetic process (9); GO:0005975 for Carbohydrate metabolic process (10); GO:0006164 and GO:0009117 for Nucleotide metabolic process (11); GO:0006094 for Gluconeogenesis (12); GO:0055076 for Transition metal ion homeostasis (13); GO:0006873 for Cellular ion homeostasis (14); GO:0007601 for Visual perception (15); GO:0007517 and GO:0090257 for Muscle tissue development (16); GO:0051216 for Cartilage development (17); GO:0042391 for Regulation of membrane potential (18); GO:0031099 for Regeneration (19).

DOI: https://doi.org/10.7554/eLife.26845.012

medaka (*Figure 5A*, compare lane 12 with lane 11), both maternal and zygotic IRE1α C156X mRNA is subjected to nonsense-mediated mRNA decay. Maternal IRE1α WT mRNA present in embryos at 0.5 dpf of IRE1α/β-DKO medaka is translated to produce IRE1α (protein). This initiates the splicing of zygotic XBP1 WT mRNA, resulting in the production of pXBP1(S) and better development of the hatching gland in IRE1α/β-DKO medaka than in XBP1-KO medaka (*Figure 3*). IRE1α (protein) translated from maternal IRE1α WT mRNA is likely to be degraded by 0.7 dpf, given that the level of spliced XBP1 mRNA was markedly decreased in embryos at 0.7 dpf of IRE1α/β-DKO medaka (*Figure 5B*, compare lane 8 with lane 7). Our subsequent crossing using male and female IRE1α/β-DKO XBP1S$^C$/+ medaka firmly supported the notion that the presence of maternal IRE1α affects development of the hatching gland (*Figure 10C and D*).

Yeast IRE1 is responsible for transcriptional induction of all UPR target genes, including various ER chaperone genes (*Travers et al., 2000*). In marked contrast, deletion of ubiquitously expressed IRE1α had no effect on transcriptional induction of ER chaperone genes in mice (*Lee et al., 2002*; *Urano et al., 2000*), and Ron and his colleagues instead showed that the JNK pathway was not activated in mouse embryonic fibroblast cells deficient in IRE1α when treated with potent ER stress inducers, such as 1 μM thapsigargin, 2.5 μg/ml tunicamycin and 10 mM dithiothreitol (*Urano et al., 2000*). Since then, IRE1α-mediated activation of the JNK pathway has been shown to be involved in ER stress-induced cell death, including the homocysteine-induced death of human vascular endothelial cells (*Zhang et al., 2001*), neuronal cell death triggered by expanded polyglutamine repeats (*Nishitoh et al., 2002*), paraquat-induced apoptosis of human neuroblastoma cells (*Yang et al., 2009*), corticotropin releasing hormone-induced neuron apoptosis (*Zhang et al., 2012*), cadmium-induced apoptosis of renal tubular cells (*Kato et al., 2013*), Japanese encephalitis virus-induced apoptosis of BHK-21 cells (*Huang et al., 2016*), and arjuric acid-induced apoptosis in non-small cell lung cancer cells (*Joo et al., 2016*).

Metazoan IRE1 initiates a highly specific splicing reaction of XBP1 mRNA to produce pXBP1(S) by cleaving two characteristic stem-loop structures in response to ER stress (*Mori, 2009*). Hollien and Weissman discovered that IRE1 also carries out degradation of a certain set of mRNA by cleaving them relatively non-specifically in Drosophila S2 cells (*Hollien and Weissman, 2006*), and later in mammalian cells (*Hollien et al., 2009*) when treated with potent ER stress inducers, such as 500 nM thapsigargin, 3 μg/ml tunicamycin and 2 mM dithiothreitol via the process termed RIDD. Since then, RIDD was shown to be involved in many phenomena, for example, maintenance of pancreatic β-cell homeostasis via degradation of insulin1 and insulin2 mRNAs on treatment of INS-1 832/13 cells with high levels of glucose (*Lipson et al., 2008*); insufficient maturation of insulin in XBP1-silenced Min6 insulinoma cells via degradation of mRNAs encoding components of the insulin secretory pathway (*Lee et al., 2011*); protection against acetaminophen-induced hepatotoxicity via degradation of Cyp1a2 and Cyp2e1 mRNAs whose translational products convert acetaminophen to hepatotoxic metabolites in adult mice with liver specific-conditional XBP1-KO (*Hur et al., 2012*); photoreceptor differentiation and rhabdomere morphogenesis in drosophila via degradation of mRNA encoding fatty acid transport protein (*Coelho et al., 2013*); induction of an inflammatory response triggered by cholera toxin via production of mRNA fragments which activate retinoic-acid inducible gene 1, leading to activation of NF-kB and interferon (*Cho et al., 2013*); and Japanese encephalitis virus-induced degradation of endogenous mRNAs in mouse neuroblastoma cells which benefits viral

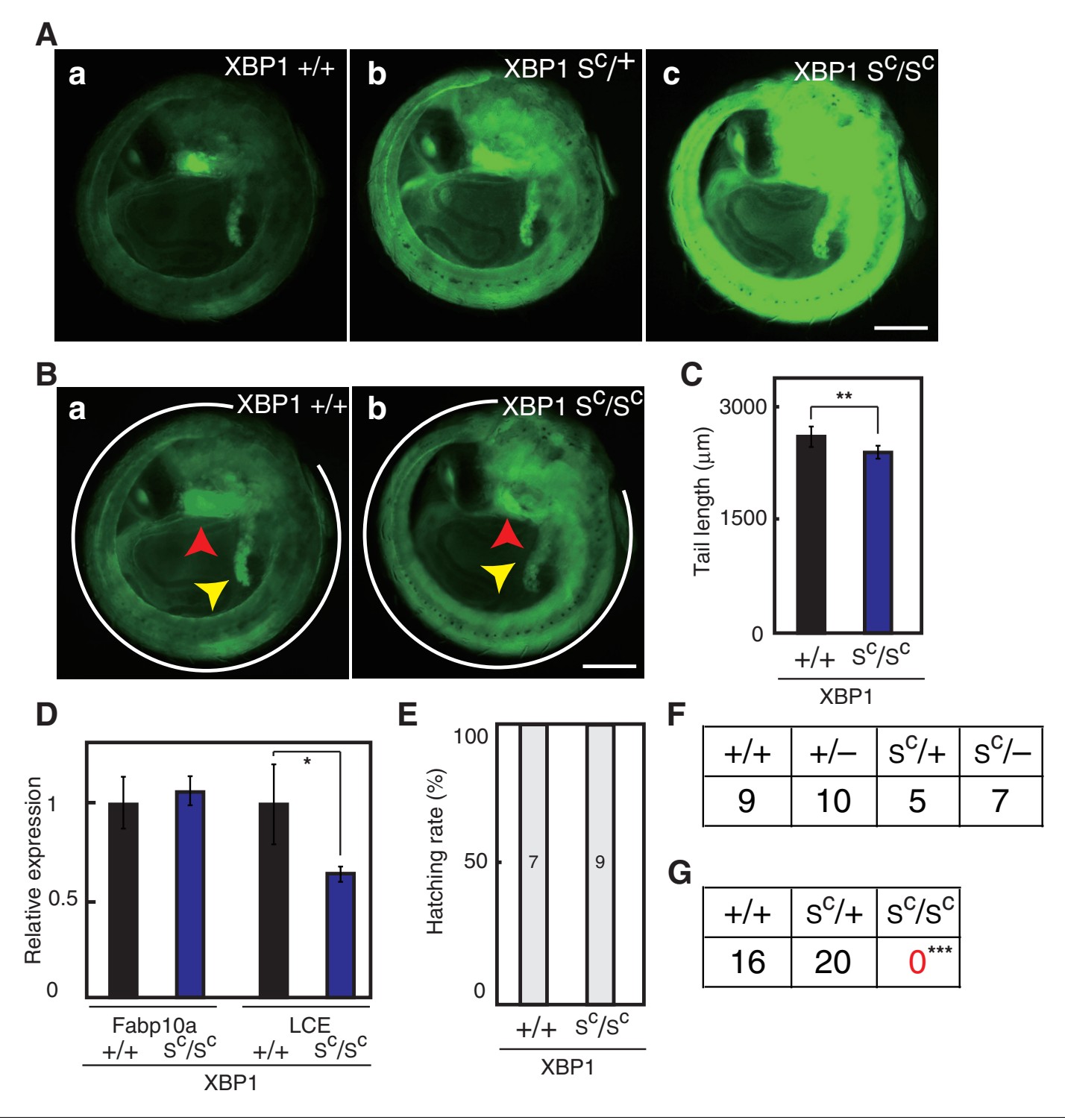

**Figure 12.** Effect of constitutive expression of pXBP1(S) from both alleles on medaka growth and development. (A) Fluorescence microscopic analysis of XBP1+/+, XBP1S$^C$/+ and XBP1S$^C$/S$^C$ embryos at 5 dpf obtained by incrossing male and female XBP1S$^C$/+ medaka, both carrying P$_{BiP}$-EGFP. Scale bar: 250 μm. (B) Longer exposure time of A(a) and shorter exposure time of A(c) for better comparison. Scale bar: 250 μm. (C) Tail lengths of the indicated genotypes were measured and are expressed as in *Figure 3A* (n > 3). (D) Levels of Fabp10a mRNA and LCE mRNA in embryos of the indicated genotypes at 5 dpf were determined and normalized with the level of β-actin mRNA, and are expressed as described in *Figure 3B* (n = 3). (E) Hatching rates of embryos of the indicated genotypes were determined and are expressed as described in *Figure 3F*. (F) Male XBP1S$^C$/+ medaka was

*Figure 12 continued on next page*

*Figure 12 continued*

crossed with female XBP1+/− and the 31 resulting fishes were genotyped 2 months after hatching. (**G**) Male and female XBP1S$^C$/+ medaka was incrossed and the 36 resulting fishes were genotyped 2 months after hatching.

DOI: https://doi.org/10.7554/eLife.26845.013

replication (*Bhattacharyya et al., 2014*). Very recent studies revealed cell-type specific activation of RIDD in response to XBP1 deletion among various type1 conventional dendritic cells for survival (*Tavernier et al., 2017*) and a role of RIDD in premature senescence of primary mouse keratinocytes following proliferation induced by oncogenic Ras (*Blazanin et al., 2017*).

Here, we investigated whether the JNK and RIDD pathways play critical roles in counteracting the physiological ER stress that occurred during growth and development of medaka fish. Because complete removal of the IRE1α/β genes from XBP1-KO medaka had no effect on the defects in the three organs (*Figure 4*), and because the defects in the three organs observed in IRE1α/β-DKO medaka were completely rescued by introduction of XBP1S$^C$ into one XBP1 allele from which fully spliced XBP1 mRNA is constitutively expressed (*Figure 9*), we concluded that the JNK and RIDD pathways are not required for normal growth and development of medaka fish. Thus, IRE1α/β-mediated splicing of XBP1 mRNA and thus production of pXBP1(S), but not of pXBP1(U) or pXBP1(AS), is sufficient to cope with the physiological ER stress that occurs during normal growth and development of medaka fish. RNA-seq analysis firmly supported this notion (*Figure 11*). In other words, such physiological ER stresses are manageable by IRE1α/β-mediated splicing of XBP1 mRNA, whose extent can be adjusted to the strength or duration of stress from the basal level to the maximal level (equivalent to the level of spliced XBP1 mRNA produced from the XBP1S$^C$ allele, see *Figure 8D*). It should be noted that morpholino-mediated knockdown experiments were employed previously to show XBP1-mediated development of the hatching gland in zebrafish (*Bennett et al., 2007*) and XBP1 and BBF2H7-mediated development of the notochord in *Xenopus* embryo (*Tanegashima et al., 2009*), with no discrimination among pXBP1(S), pXBP1(U) and pXBP1(AS). It appears that activation of the JNK and RIDD pathways is triggered by stimuli which evoke greater or different types of ER stress than those stresses encountered during normal development.

XBP1-KO causes embryonic lethality in mice due to the failure of liver development (*Reimold et al., 2000*). In contrast, liver specific-conditional XBP1-KO in adult mice has almost no effect on the growth of mice or secretion of proteins from hepatocytes (*Lee et al., 2008*), probably due to compensation by the ATF6 pathway, which is responsible for transcriptional induction of ER chaperone genes in response to ER stress in both mice and medaka, but not in non-vertebrates (*Ishikawa et al., 2013*; *Yamamoto et al., 2007*). Interestingly, liver specific-conditional XBP1-KO mice showed hypolipidemia and XBP1 was found to regulate transcription of non-canonical UPR target genes which encode four enzymes involved in lipidogenesis (*Lee et al., 2008*). Furthermore, the absence of XBP1 in mouse liver hyperactivated IRE1α, and hypolipidemia is accordingly accelerated by RIDD of mRNAs encoding proteins involved in lipogenesis and lipoprotein metabolism (*So et al., 2012*),

Based on these results, we consider that the protein quality control system is maintained by the IRE1-XBP1 and ATF6 branches of the UPR and not by the RIDD pathway at the whole-body level of mice and medaka. Consistent with this idea, hyperactivation of IRE1α occurring in cartilage-specific conditional XBP1-KO mice was not sufficient for activation of RIDD, and synthesis and secretion of type II and type X collagen was unaffected by specific deletion of XBP1 in cartilage (*Cameron et al., 2015*). The RIDD pathway may have more specific functions in vertebrates, for example regulation of lipid metabolism working together with non-canonical function of XBP1, and protection against toxins, such as acetaminophen and cholera toxin. It is important to discriminate the protein quality control system from other events that occurs in the ER in vertebrates.

Very recently, H. D. Ryoo and his colleagues investigated the requirement of the IRE1-XBP1 branch in coping with physiological ER stress which occurs during embryonic development of *Drosophila melanogaster* and found that IRE1-mediated signaling independent of XBP1 mRNA splicing, particularly RIDD, is also functionally important in this organism (*Huang et al., 2017*). We speculate that the advent of functional ATF6 might have mitigated the importance of RIDD in vertebrates, because ATF6-mediated rapid induction of various ER chaperones for refolding of unfolded/

misfolded proteins appears to be a more sophisticated way of coping with ER stress than RIDD-mediated relatively nonspecific degradation of various mRNAs, from which no functional proteins would be produced.

## Materials and methods

### Fish

Medaka southern strain cab was used as wild-type fish. Fish were maintained in a recirculating system with a 14:10 hr light:dark cycle at 27.5°C. All experiments were performed in accordance with the guidelines and regulations established by the Animal Research Committee of Kyoto University (approval number: H2819). EGFP imaging was performed under a fluorescence stereomicroscope (Leica M205FA; Wetzlar, Germany) using a GFP3 filter (470/40 nm excitation filter and 525/50 nm barrier filter) with a camera (Leica DFX310FX) and acquisition software (Leica las AF).

A strain carrying $P_{BiP}$-EGFP was described previously (*Ishikawa et al., 2011*). A strain carrying $P_{LCE}$-EGFP was obtained from the National Institute for Basic Biology.

Statistical analysis was conducted using Student's t-test. *p<0.05, **p<0.01, and ***p<0.001 for all figures.

### TILLING method

The missense mutation of IRE1α (C156X) or IRE1β (Y164X) was identified from a library of 5760 mutated male fishes of N1 (*Taniguchi et al., 2006*) by subjecting PCR fragments amplified from this library using the primer sets 5'-TTTGTCGCTGTGCTTATTAC-3' and 5'-TGCTGTGGTTTGGGTTT-3' for IRE1α as well as 5'-TGCACGCCTTCTTTTCC-3' and 5'-TCCCAGCCAGAAAATGTT-3' for IRE1β to high-resolution melting curve analysis (*Ishikawa et al., 2010*) and subsequent sequencing. All IRE1-KO experiments were conducted by crossing IRE1α (N10) or IRE1β (N6) homozygotes/heterozygotes.

### TALEN and CRISPR-Cas9 methods

To construct TALEN, TAL repeats were assembled by the modified Golden Gate assembly method (*Sakuma et al., 2013*).

To construct sgRNA vectors for CRISPR-Cas9, two oligonucleotides corresponding to the 5'-side of exon 4 of the XBP1 gene (5'-taggCTGCGGACTCAGAAGACC-3' and 5'-aaacGGTCTTCTGAGTCCGCAG-3'), and two oligonucleotides corresponding to the 3'-side of exon 4 of the XBP1 gene (5'-taggTGCCTCCGCAGCAGGTGC-3' and 5'-aaacGCACCTGCTGCGGAGGCA-3') were annealed and ligated into BsaI-digested DR274 vector (Addgene). sgRNA vectors were linearized with DraI, purified by phenol chloroform extraction, and used as template to synthesize sgRNAs using T7 RNA polymerase.

TALEN and Cas9(D10A) expressing vectors were linearized with NotI, purified by phenol chloroform extraction, and used as template to synthesize capped mRNAs using the Message mMachine SP6Kit (Life Technologies, Gaithersburg, MD).

Synthesized RNAs were purified by RNeasy MinElute (Qiagen, Germany) and microinjected into one-cell stage embryos at the concentration of 50 ng/μl for both left and right TALEN, 100 ng/μl for Cas9(D10A), and 25 ng/μl for both sense and antisense strand sgRNA of Cas9. Injection was performed as described previously (*Ishikawa et al., 2011*).

### Genotyping

Embryos or hatched fishes were suspended in 50 μl of lysis buffer (10 mM NaOH and 0.2 mM EDTA), boiled for 10 min, then neutralized by the addition of 50 μl of 40 mM Tris/HCl, pH8.0. The DNA fragment containing the mutation site of XBP1 Δ8, Δ4, Δ15, Δ26, IRE1α (C156X) or IRE1β (Y164X) was amplified by PCR directly from lysates using the primers 5'-TTTGAATTCTCATGGTGGTGGTCGCAACGG-3' and 5'-CTGGCCTCATTTATTTCCGAAGT-3' for XBP1 Δ8, 5'-GACAGAAAATGAGGAACTGAGACAGAGAC-3' and 5'-GACTTGAGAAACAGCTCTGGGTCAAGGAT-3' for XBP1 Δ4, Δ15 and Δ26, 5'-TTACCAGGGAAAAAGCAGGACTTGTGGTATGTGGTGGACCTGTTGACTGGTGAGAAACAGCAGACCTTGA-3' and 5'-TACGGCCGTGTCAAGCTTGTGTCAAGATTA-3' for IRE1α

(C156X), and 5′-AGGTCGATGCTTTGACTGA-3′ and 5′-GTGTGTGTGTGCATGGACATA-3′ for IRE1β (Y164X).

Amplified PCR fragments were subjected to restriction enzyme digestion {ApeKI for XBP1 Δ8, Cac8I for Δ4, Hpy188I for Δ15, AflIII for Δ26, and Hpy188I for IRE1α (C156X)} and then electrophoresed. Amplified fragments for IRE1β were directly sequenced.

## RT-PCR and quantitative RT-PCR

Total RNA was extracted from embryos at the indicated dpf by the acid guanidinium/phenol/chloroform method using Isogen (Nippon Gene, Tokyo, Japan). RT-PCR analysis to detect ER stress-induced splicing of XBP1 mRNA was carried out as described previously (*Ishikawa et al., 2011*). Quantitative RT-PCR analysis was carried out as described previously (*Ishikawa et al., 2013*) using the SYBR Green method (Applied Biosystems) and a pair of primers, namely 5′-CGGTATCCATGA-GACCACCT-3′ and 5′-AGCACAGTGTTGGCGTACAG-3′ for β-actin mRNA, 5′-TCCGCTGAGTCC-taAGCAGG-3′ and 5′-GTCTGAAGAGTAAGGACTATCTGT-3′ for spliced XBP1 mRNA, 5′-GCTCGG TCAGAGACCTGCT-3′ and 5′-CCTGTACCTCTGCAGGCAAT-3′ for IRE1α mRNA, 5′-GGTC TGCAAAACTGGCAAAT-3′ and 5′-TTGTTGAGCCAACGGTCATA-3′ for Fabp10a mRNA, 5′-TGAC TATGGCTCCATCATGC-3′ and 5′-TTAGGGATGGGGGGTTATGGT-3′ for LCE mRNA, 5′-TGGTCTG TGGAGACAGAGGAT-3′ and 5′-CCATCTTTGGGGAGAGGAG-3′ for Ins1 mRNA, 5′-AGCACTGA-CAACTACCAGAGT-3′ and 5′-TCTTTGCATCCACACCGTCA-3′ for Fabp2a mRNA, 5′-GACTACTGA TGACCCAGGCAG-3′ and 5′-GGGTCAATGGGTGCGAGAG-3′ for Mcmlc2 mRNA, 5′-CTTC TCCGAGGCTCTGAACG-3′ and 5′-ACAATGCTGGACAGGCAGTC-3′ for MyoD mRNA, 5′-AGCG TACAGATTTGCCCAGC-3′ and 5′-GTGGTAGGGGTCTGCTGTTG-3′ for NeuN mRNA. The absolute expression levels of spliced XBP1 mRNA and β-actin mRNA were determined using 600, 6000, and 60,000 molecules of plasmid carrying the XBP1 gene, and 60,000, 600,000, and 6,000,000 molecules of plasmid carrying the β-actin gene, respectively.

## Construction of plasmids

Recombinant DNA techniques were performed according to standard procedures (*Sambrook et al., 1989*) and the integrity of all constructed plasmids was confirmed by extensive sequencing analyses. Plasmids to express various forms of XBP1 were constructed as described previously (*Ishikawa et al., 2011*). Medaka IRE1α cDNA obtained previously (*Ishikawa et al., 2011*) was subcloned into pcDNA to create pcDNA-medaka IRE1α.

## mRNA purification for RNA-seq

15 μl of Dynabeads Oligo (dT)25 (ThermoFisher Scientific) was washed with 50 μl of 2 × binding buffer (40 mM Tris-HCl, pH7.6, 2 M LiCl, 4 mM EDTA) twice using Magna Stand for 0.2 ml PCR Tube (FastGene) and resuspended in 30 μl of 2 × binding buffer. 1 μg of total RNA in 30 μl of distilled water was denatured at 65°C for 2 min and then immediately chilled on ice. 30 μl of the washed Dynabeads Oligo (dT)25 was then added. The mixture was mixed well and incubated at room temperature for 10 min. The mixture was washed with 70 μl of washing buffer (10 mM Tris-HCl, pH7.6, 0.15 M LiCl, 1 mM EDTA) twice using the Magna Stand. The RNA was eluted in 30 μl of TE at 80°C for 5 min and then immediately chilled on ice. 30 μl of 2 × binding buffer was added and incubated at room temperature for 10 min. Again, the mixture was washed with 70 μl of washing buffer twice using the Magna Stand. The RNA was eluted in 14 μl of distilled water at 80°C for 2 min, then immediately placed on the magnet and collected.

## RNA-seq library preparation and sequencing

5 μl of purified mRNA obtained in the above section was mixed with 4 μl of 5 × SuperScript IV buffer (Invitrogen) and 1 μl of 100 mM DTT (Invitrogen). Fragmentation of mRNA was carried out at 94°C for 4.5 min and immediately cooled down on ice. 0.6 μl of 100 μM random primer $(N)_6$ (TaKaRa) and 0.9 μl of distilled water were then added to the mixture. The mixture was incubated at 50°C for 5 min and immediately chilled on ice to relax the secondary structures of the RNA. The fragmented RNA with the random hexamer and the reverse transcription master mix {1 μl of 100 mM DTT, 0.4 μl of dNTP (25 mM each) (Promega); 0.1 μl of SuperScript IV (Invitrogen); 0.2 μl of Actinomycin D (1000 ng/μl) (Nacalai Tesque); and 5.9 μl of distilled water} was mixed. For the reverse transcription step,

the mixture was incubated at 25°C for 10 min, followed by 50 min at 50°C. SuperScript IV was inactivated by heating the mixture at 75°C for 15 min. 24 µl of AMPure XP (Beckman Coulter) and 12 µl of 99.5% ethanol were added, and purification was performed according to the manufacturer's manual. The traverse transcription product was eluted with 10 µl of distilled water. The purified DNA/RNA hybrid solution without beads and the second strand synthesis master mix (2 µl of 10 × Blue Buffer (Enzymatics), 1 µl of dUTP/NTP mix (Fermentas), 0.5 µl of 100 mM DTT, 0.5 µl of RNase H (Enzymatics), 1 µl of DNA polymerase I (Enzymatics), and 5 µl of distilled water) were mixed. The mixture was incubated at 16°C for 4 hr. Purification was then performed with 24 µl of AMPure XP according to the manufacturer's manual. The purified dsDNA was eluted with 10 µl of distilled water. 5 µl of the dsDNA solution was used for the following step. End-repair, A-tailing and adapter ligation were carried out using a Kapa Hyper prep kit (Kapa Biosystems) with 1/10 × volume of the solutions according to the manufacturer's manual. 1 µl of 0.1 µM Y-shape adapter (*Nagano et al., 2015*) was used in the adapter ligation step for 15 min. Size selection of the ligation product was then performed with 5.5 µl of AMPure XP. The purified dsDNA was eluted using 10 µl of distilled water. The second round of size selection was performed with 10 µl of AMPure XP. The size-selected ligation product was eluted with 15 µl of 10 mM Tris–HCl, pH 8.0. 1 µl of uracil DNA glycosylase (UDG) (Enzymatics) was added to the size-selected ligation product. The mixture was incubated at 37°C for 30 min to exclude the second-strand DNA. For library amplification, 2 µl of the UDG-digested DNA, 1 µl of 2.5 µM index primer (*Nagano et al., 2015*), 1 µl of 10 µM universal primer (*Nagano et al., 2015*), 0.5 µl of distilled water and 5 µl of Kapa HiFi HotStart ReadyMix (2×) (Kapa Biosystems) were mixed. DNA fragments with the adapters and an index sequence were amplified using a thermal cycler with the following program: denature at 94°C for 2 min, 18 cycles at 98°C for 10 s, 65°C for 30 s, 72°C for 30 s as an amplification step, and 72°C for 5 min for the final extension. Two rounds of size selection were then performed to remove adapter dimer with equal volume of AMPure XP to the library solution. The purified library was eluted with 10 µl of distilled water. 1 µl of the purified library was used for electrophoresis using an Agilent High Sensitivity DNA kit (Agilent Technologies) to evaluate quality. Sequencing using Hiseq 2500 (Illumina, Inc.) was carried out by Macrogen Co.

## Data analysis

All reads were trimmed with trimmomatic (version 0.3.3) (*Bolger et al., 2014*) with the following parameters. TOPHRED33 ILLUMINACLIP: TruSeq3-SE.fa:2:30:10 LEADING:19 TRAILING:19 SLIDINGWINDOW:30:20 AVGQUAL:20MINLEN:40'. The trimmed reads were then mapped onto the medaka cDNA reference sequences (Oryzias_latipes.MEDAKA1.cdna.all.fa.gz) downloaded from the ensemblgenome database (http://www.ensembl.org) with RSEM (version 1.3.0) (*Li and Dewey, 2011*) using bowtie (version 1.1.2) (*Langmead et al., 2009*) with default parameters.

Differentially expressed gene analysis was performed using the estimated read counts with the TCC package using edgeR (version 1.14.0) on R (version 3.3.3) as described by Sun et. al. (*Sun et al., 2013*). GO enrichment analysis was performed using GO of homologous human genes annotated in the ensembl database with clusterProfiler package (version 3.2.14) and org.Hs.eg.db annotation package (ver 3.4.0) on R (version 3.3.3).

## Cell culture

HCT116 cells (ATCC CCL-247) were cultured in Dulbecco's modified Eagle's medium (glucose 4.5 g/liter) supplemented with 10% fetal bovine serum, 2 mM glutamine, and antibiotics (100 U/ml penicillin and 100 µg/ml streptomycin) at 37°C in a humidified 5% $CO_2$/95% air atmosphere, and transfected using Peimax.

## Immunological techniques

Immunoblotting analysis was carried out according to the standard procedure (*Sambrook et al., 1989*) as described previously (*Ishikawa et al., 2011*). Chemiluminescence obtained using Western Blotting Luminol Reagent (Santa Cruz Biotechnology) was detected using an LAS-3000mini Lumino-Image analyzer (Fuji Film). Rabbit anti-c-myc antibody (9E10) was obtained from Wako.

## Acknowledgements

We thank Ms. Kaoru Miyagawa and Ms. Yuki Okada for their technical and secretarial assistance. We are grateful to Dr. Kiyoshi Naruse at the National Institute for Basic Biology for providing NBRP medaka materials. This work was financially supported in part by grants from the Ministry of Education, Culture, Sports, Science and Technology of Japan (26291040 and 17H01432 to KM, 15K18529 and 17K15116 to TI) and supported by Collaborative Research Program of National Institute for Basic Biology (14–387 to TI).

## Additional information

### Funding

| Funder | Grant reference number | Author |
|---|---|---|
| Ministry of Education, Culture, Sports, Science, and Technology | 26291040 | Kazutoshi Mori |
| Ministry of Education, Culture, Sports, Science, and Technology | 15K18529 | Tokiro Ishikawa |
| Ministry of Education, Culture, Sports, Science, and Technology | 17H01432 | Kazutoshi Mori |
| Ministry of Education, Culture, Sports, Science, and Technology | 17K15116 | Tokiro Ishikawa |

The funders had no role in study design, data collection and interpretation, or the decision to submit the work for publication.

### Author contributions

Tokiro Ishikawa, Conceptualization, Formal analysis; Makoto Kashima, Atsushi J Nagano, Tomoko Ishikawa-Fujiwara, Yasuhiro Kamei, Takeshi Todo, Formal analysis, Methodology; Kazutoshi Mori, Supervision, Writing—original draft

### Author ORCIDs

Tokiro Ishikawa http://orcid.org/0000-0003-1718-6764
Kazutoshi Mori http://orcid.org/0000-0001-7378-4019

### Ethics

Animal experimentation: All experiments were performed in accordance with the guidelines and regulations established by the Animal Research Committee of Kyoto University (approval number: H2819).

### Decision letter and Author response

Decision letter https://doi.org/10.7554/eLife.26845.017
Author response https://doi.org/10.7554/eLife.26845.018

## Additional files

### Supplementary files

• Transparent reporting form
DOI: https://doi.org/10.7554/eLife.26845.014

### Major datasets

The following dataset was generated:

| Author(s) | Year | Dataset title | Dataset URL | Database, license, and accessibility information |
|---|---|---|---|---|
| Ishikawa T, Mori K | 2017 | tokiro-0001 | http://trace.ddbj.nig.ac.jp/DRASearch/submission?acc=DRA006141 | Publicly available at the DDBJ (accession no. DRA006141) |

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
