## [Decision Letter]

Thank you for submitting your article "UPR sensor IRE1-mediated Signaling Independent of XBP1 mRNA Splicing Is Not Required for Growth and Development of Medaka" for consideration by *eLife*. Your article has been favorably evaluated by Didier Stainier (Senior Editor) and three reviewers, one of whom is a member of our Board of Reviewing Editors. The reviewers have opted to remain anonymous.

The reviewers have discussed the reviews with one another and the Reviewing Editor has drafted this decision to help you prepare a revised submission.

Summary:

In this study, the authors have used the vertebrate model system of the medaka fish to systematically investigate the different functional outputs of the Ire1 signaling pathway on normal development. Ire1 is the most widely conserved ER stress sensor that communicates misfolding stress to the nucleus via splicing of Xbp1. In addition, signaling and mRNA degradation functions of Ire1 have been described and extensively studied. The relative contributions of the canonical Xbp1-mediated signaling versus the other Ire1 outputs to normal development or to different physiologic states is not clear. This study provides clear genetic evidence that development and early life of medaka can proceed normally with only the Xbp1 output, suggesting that the other signaling and mRNA degradation outputs are only needed under certain other conditions that remain to be explored.

In general, the study is quite straightforward in presenting the two central results: (i) knockout of Ire1 (both isoforms) can be fully rescued with constitutive modest level spliced Xbp1 expression; (ii) the Ire1/Xbp1 double knockout is no more severe than the Ire1 knockout. The conclusion is that other nuances of this system, such as non-Xbp1 mediated Ire1 outputs, a role for unspliced Xpb1, and a role for alternatively spliced Xbp1, are apparently dispensable in this organism for normal early development. In addition, it seems that constitutive Xbp1 can replace the stress-responsive regulation that would normally occur with an intact Ire1-Xbp1 axis.

The referees agreed that the above conclusion was generally well supported (with some potential improvements needed; see below). However, we initially disagreed somewhat regarding the overall impact of these observations. The major concern was that the results do not really provide new insights into which Ire1/Xbp1 functions make it essential, nor does is really speak to whether RIDD or JNK signaling even occurs during early development. In this sense, the function(s) of these latter two outputs of Ire1 activation are not illuminated beyond showing that they are non-essential for medaka development. While this is certainly a valid criticism, additional discussion among the referees led to the consensus view that the study is nevertheless important for this field and beyond. The reason is that the findings were felt to provide an important baseline for future study, and clarify exactly which aspect of the various putative Ire1 functions is actually central to normal physiology and which are more modulatory or even unnecessary. The animal models described here should also be useful in different experimental paradigms to probe the situations when these other outputs become more important. For these reasons, we concluded that if the study can indeed provide definitive evidence that the Xbp1s output is the only essential function of Ire1 activation, it will help focus the UPR field and represent an important contribution suitable for *eLife*.

Essential revisions:

The major issues revolve around a more thorough characterization of the genotypes and phenotypes of the animals to conclusively document their mutants, and that Xbp1s fully rescues the Ire1 double knockout. As currently presented, the relatively superficial and qualitative morphological analysis was judged to be too blunt, and some confusion remains about the putative maternal effect that seems to cause different phenotypes in the Ire1 versus Xbp1 knockouts.

1) The alleles used have not been fully characterized, and stronger evidence is needed to convincingly document that the Ire1 alleles are null, and that Xbp1 alleles are behaving as expected. Blotting to assess protein levels generated from the mutant alleles is one option, but we acknowledge that suitable reagents may not be available in this system. Alternatively, analysis of the effect of each mutant on the targeted transcript (e.g., Northerns, qPCR, and/or direct sequencing) would help substantiate the authors' claims. This is a concern because the point mutants in IRE1a and IRE1b generated by TILLING mutants may not generate the truncated protein as expected, and if the alleles are not truly null, the conclusions are weakened. Similarly, it is important to verify that the Xbp1 alleles are as expected at the mRNA level.

2) There is inadequate phenotypic analysis of the mutants. The relatively qualitative description of the organs that are proposed to be affected (tail, hatching gland, liver) does not provide enough information to be fully convinced. Some type of unbiased quantitative assessment with statistical analysis would be strongly preferred, especially for the critical result showing that the Xbp1s fully rescues the Ire1 double knockout. A complementary phenotypic analysis would be a transcriptomic comparison between the wild type, Ire1 double knockout, and Xbp1s rescue strains to how complete the rescue really is. We can appreciate that this might be judged beyond the scope of this study, but such molecular analysis would significantly strengthen the claim of full rescue based on morphological criteria.

3) The final concern relates to the phenotype of IRE1a/b DKO fish. The authors argue in Figure 4 that there is a maternal effect in terms of production of IRE1 protein that is sufficient to splice Xbp1 RNA in early embryos. While their data are consistent with this possibility, they cannot conclude a maternal effect is present without systematically analyzing the phenotypes in crosses where the maternal genotype is varied. In addition, the authors seem a bit coy about the phenotype of IRE1a/b DKO animals. The table in Figure 7 implies that such animals are not viable, yet clearly they are, as they are used in other crosses (e.g., as the females in Figure 7). This discrepancy might be because of the presumed maternal effect but again, the paper seems to report no experiment done for the express purpose of testing maternal genotype. The other reason this matters is that, if IRE1a/b DKO animals are in fact viable, it is a bit difficult to reconcile that observation with the much more severe phenotype in Xbp1 KO animals. The authors need to clarify the phenotypes of animals of various genotypes, and if they wish to make strong claims about a putative maternal effect explaining differences between the Ire1 DKO and Xbp1 knockout, then the idea needs to be more directly tested.

---

## [Author Response]

Essential revisions:The major issues revolve around a more thorough characterization of the genotypes and phenotypes of the animals to conclusively document their mutants, and that Xbp1s fully rescues the Ire1 double knockout. As currently presented, the relatively superficial and qualitative morphological analysis was judged to be too blunt, and some confusion remains about the putative maternal effect that seems to cause different phenotypes in the Ire1 versus Xbp1 knockouts.1) The alleles used have not been fully characterized, and stronger evidence is needed to convincingly document that the Ire1 alleles are null, and that Xbp1 alleles are behaving as expected. Blotting to assess protein levels generated from the mutant alleles is one option, but we acknowledge that suitable reagents may not be available in this system. Alternatively, analysis of the effect of each mutant on the targeted transcript (e.g., Northerns, qPCR, and/or direct sequencing) would help substantiate the authors' claims. This is a concern because the point mutants in IRE1a and IRE1b generated by TILLING mutants may not generate the truncated protein as expected, and if the alleles are not truly null, the conclusions are weakened. Similarly, it is important to verify that the Xbp1 alleles are as expected at the mRNA level.

We confirmed the presence of the expected mutation in IRE1α mRNA or IRE1β mRNA expressed in respective KO embryo at 5 dpf by direct sequencing of RT-PCR products (subsection “IRE1α/β-double knockout (DKO) is detrimental to growth and development of medaka”, first paragraph and Figure 1).

We confirmed the presence of the expected deletion in XBP1 mRNA expressed in KO embryo at 5 dpf by direct sequencing of RT-PCR products (subsection “XBP1-KO medaka exhibit a more severe phenotype than IRE1α/β-DKO medaka in the hatching gland”, first paragraph and Figure 2).

We confirmed the presence of the expected deletion in respective XBP1 mRNA expressed in embryo at 5 dpf by direct sequencing of RT-PCR products (Figure 7), namely Δ4, Δ15, and Δ26 in XBP1 cDNA prepared from Δ4/Δ4 embryo (see Figure 8), from Δ15/Δ8 (U^c^/-) embryo (see Figure 8; note that Δ8 XBP1 mRNA is unstable, probably due to nonsense-mediated mRNA decay), and from S^C^/S^C^ embryo (see Figure 12Ac), respectively (subsection “Constitutive expression of the spliced form of XBP1 fully rescues the defects observed in XBP1-KO medaka”, last paragraph).

In addition, we showed that mutant XBP1 proteins behaved as expected by immunoblotting analysis of HCT116 cells into which plasmid to express each XBP1 mutant with or without plasmid to express medaka IRE1α (subsection “Constitutive expression of the spliced form of XBP1 fully rescues the defects observed in XBP1-KO medaka”, fourth paragraph and Figure 7).

2) There is inadequate phenotypic analysis of the mutants. The relatively qualitative description of the organs that are proposed to be affected (tail, hatching gland, liver) does not provide enough information to be fully convinced. Some type of unbiased quantitative assessment with statistical analysis would be strongly preferred, especially for the critical result showing that the Xbp1s fully rescues the Ire1 double knockout. A complementary phenotypic analysis would be a transcriptomic comparison between the wild type, Ire1 double knockout, and Xbp1s rescue strains to how complete the rescue really is. We can appreciate that this might be judged beyond the scope of this study, but such molecular analysis would significantly strengthen the claim of full rescue based on morphological criteria.

We have conducted RNA-seq analysis of transcripts expressed in embryos of IRE1β-KO, IRE1α/β-DKO XBP1+/+ and IRE1α/β-DKO XBP1 S^C^/+ and showed that the observed full rescue of the defects in IRE1α/β-DKO by introduction of XBP1S^C^ is ascribable to transcriptional activity of XBP1S^C^ (subsection “Constitutive expression of the spliced form of XBP1 fully rescues the defects observed in IRE1α/β-DKO medaka”, fourth paragraph and Figure 11).

3) The final concern relates to the phenotype of IRE1a/b DKO fish. The authors argue in Figure 4 that there is a maternal effect in terms of production of IRE1 protein that is sufficient to splice Xbp1 RNA in early embryos. While their data are consistent with this possibility, they cannot conclude a maternal effect is present without systematically analyzing the phenotypes in crosses where the maternal genotype is varied. In addition, the authors seem a bit coy about the phenotype of IRE1a/b DKO animals. The table in Figure 7 implies that such animals are not viable, yet clearly they are, as they are used in other crosses (e.g., as the females in Figure 7). This discrepancy might be because of the presumed maternal effect but again, the paper seems to report no experiment done for the express purpose of testing maternal genotype. The other reason this matters is that, if IRE1a/b DKO animals are in fact viable, it is a bit difficult to reconcile that observation with the much more severe phenotype in Xbp1 KO animals. The authors need to clarify the phenotypes of animals of various genotypes, and if they wish to make strong claims about a putative maternal effect explaining differences between the Ire1 DKO and Xbp1 knockout, then the idea needs to be more directly tested.

Our previous description on the phenotype of α/β-DKO medaka was not enough.

In the revised manuscript, we have first described that “It should be noted that all IRE1α/β-DKO medaka die within two weeks after fertilization regardless of their ability or inability to hatch (data not shown here, see Figure 10; medaka usually hatch at 7 dpf and become fertile two months after hatching). Thus, complete absence of IRE1 functions is detrimental to the growth and development of medaka” (subsection “IRE1α/β-double knockout (DKO) is detrimental to growth and development of medaka”).

We have then described that “Because IRE1α/β-DKO medaka die within two weeks after fertilization (see Figure 10) and therefore cannot be used for mating, we determined whether IRE1α/β-DKO medaka become fertile if the Δ26 mutation (XBP1S^C^) is introduced into one allele of the XBP1 locus. […] These female IRE1α-/- IRE1β-/- XBP1S^C^/+ medaka were then crossed with male IRE1α+/- IRE1β-/- XBP1+/+ medaka, resulting in the production of embryos with 4 different genotypes, as shown in Figure 9”.

We have finally described that “Subsequent crossing confirmed that IRE1α/β-DKO XBP1S^C^/+ medaka hatched and were alive for 2 months, unlike IRE1α/β-DKO XBP1+/+ medaka (Figure 10). […] These results provide clear evidence for the maternal effect of IRE1α on development of the hatching gland”